# The unprecedented 2017–2018 stratospheric smoke event: Decay phase and aerosol properties observed with EARLINET

Holger Baars[1], Albert Ansmann[1], Kevin Ohneiser[1], Moritz Haarig[1], Ronny Engelmann[1], Dietrich Althausen[1], Ingrid Hanssen[2], Michael Gausa[2], Aleksander Pietruczuk[3], Artur Szkop[3], Iwona S. Stachlewska[4], Dongxiang Wang[4], Jens Reichardt[5], Annett Skupin[1], Ina Mattis[6], Thomas Trickl[7], Hannes Vogelmann[7], Francisco Navas-Guzmán[8], Alexander Haefele[8], Karen Acheson[9], Albert A. Ruth[9], Boyan Tatarov[10], Detlef Müller[10], Qiaoyun Hu[11], Thierry Podvin[11], Philippe Goloub[11], Igor Vesselovski[12], Christophe Pietras[13], Martial Haeffelin[13], Patrick Fréville[14], Michaël Sicard[15,16], Adolfo Comerón[15], Alfonso Javier Fernández García[17], Francisco Molero Menéndez[17], Carmen Córdoba-Jabonero[18], Juan Luis Guerrero-Rascado[19], Lucas Alados-Arboledas[19], Daniele Bortoli[20,21], Maria João Costa[20], Davide Dionisi[22], Gian Luigi Liberti[22], Xuan Wang[23], Alessia Sannino[24], Nikolaos Papagiannopoulos[25], Antonella Boselli[25], Lucia Mona[25], Giuseppe D'Amico[25], Salvatore Romano[26], Maria Rita Perrone[26], Livio Belegante[27], Doina Nicolae[27], Ivan Grigorov[28], Anna Gialitaki[29], Vassilis Amiridis[29], Ourania Soupiona[30], Alexandros Papayannis[30], Rodanthi-Elisaveth Mamouri[31], Argyro Nisantzi[31], Birgit Heese[1], Julian Hofer[1], Yoav Y. Schechner[32], Ulla Wandinger[1], and Gelsomina Pappalardo[25]

[1]Leibniz Institute for Tropospheric Research, Leipzig, Germany
[2]Andøya Space Center, Andenes, Norway
[3]Institute of Geophysics, Polish Academy of Sciences, Warsaw, Poland
[4]Faculty of Physics, University of Warsaw, Warsaw, Poland
[5]The Lindenberg Meteorological Observatory, Deutscher Wetterdienst, Tauche, Germany
[6]Meteorological Observatory Hohenpeissenberg, Deutscher Wetterdienst, Hohenpeissenberg, Germany
[7]IMK-IFU, Karlsruhe Institute of Technology, Garmisch-Partenkirchen, Germany
[8]Federal Office of Meteorology and Climatology, MeteoSwiss, Payerne, Switzerland
[9]Physics Department & Environmental Research Institute, University College Cork, Cork, Ireland
[10]School of Physics, Astronomy and Mathematics, University of Hertfordshire, Hatfield, United Kingdom
[11]LOA, Université de Lille, Lille, France
[12]Physics Instrumentation Center of General Physics Institute, Moscow, Russia
[13]Laboratoire Meteorologie Dinamique, École Polytechnique, Palaiseau, France
[14]Observatoire de Physique du Globe, Laboratoire de Météorologie Physique, Clermont-Ferrand, France
[15]CommSensLab, Dept. of Signal Theory and Communications, Universitat Politècnica de Catalunya, Barcelona, Spain
[16]CTE-CRAE/IEEC, Universitat Politècnica de Catalunya, Barcelona, Spain
[17]Centro de Investigaciones Energéticas, Medioambientales y Tecnológicas, Department of Environment, Madrid, Spain
[18]Instituto Nacional de Tècnica Aeroespacial, Atmospheric Research and Instrument. Branch, El Arenosillo/Huelva, Spain
[19]Andalusian Institute for Earth System Research and University of Granada, Granada, Spain
[20]Instituto Ciências da Terra, Universidade de Évora, Evora, Portugal
[21]Departamento de Física, Universidade de Évora, Evora, Portugal
[22]Consiglio Nazionale delle Ricerche, Istituto di Scienze Marine, Rome-Tor Vergata, Italy
[23]Consiglio Nazionale delle Ricerche, Istituto Superconduttori, Materiali Innovativi e Dispositivi, Naples, Italy
[24]Dipartimento di Fisica, Università degli studi di Napoli Federico II, Naples, Italy
[25]Consiglio Nazionale delle Ricerche, Istituto di Metodologie per l'Analisi Ambientale, Potenza, Italy
[26]Consorzio Nazionale Interuniversitario per le Scienze Fisiche della Materia and Università del Salento, Lecce, Italy

[27]National Institute of Research and Development for Optoelectronics, Magurele, Ilfov, Romania

[28]Institute of Electronics, Bulgarian Academy of Sciences, Sofia, Bulgaria

[29]IAASARS, National Observatory of Athens, Athens, Greece

[30]Laser Remote Sensing Unit (LRSU), Physics Department, National Technical University of Athens, Zografou, Greece

[31]Department of Civil Engineering and Geomatics, Cyprus University of Technology, Limassol, Cyprus

[32]Viterbi Faculty of Electrical Engineering, Technion - Israel Institute of Technology, Haifa, Israel

*Correspondence to:* H. Baars

(baars@tropos.de)

**Abstract.** Six months of stratospheric aerosol observations with the European Aerosol Research Lidar Network (EARLINET) from August 2017 to January 2018 are presented. The decay phase of an unprecedented, record-breaking stratospheric perturbation caused by wild fire smoke is reported and discussed in terms of geometrical, optical, and microphysical aerosol properties. Enormous amounts of smoke were injected into the upper troposphere and lower stratosphere over fire areas in western Canada on 12 August 2017 during strong thunderstorm-pyrocumulonimbus activity. The stratospheric fire plumes spread over the entire northern hemisphere in the following weeks and months. Twenty-eight European lidar stations from northern Norway to southern Portugal and the Eastern Mediterranean monitored the strong stratospheric perturbation on a continental scale. The main smoke layer (over central, western, southern, and eastern Europe) was found between 15 and 20 km height since September 2017 (about two weeks after entering the stratosphere). Thin layers of smoke were detected up to 22-23 km height. The stratospheric aerosol optical thickness at 532 nm decreased from values >0.25 on 21-23 August 2017 to 0.005-0.03 until 5-10 September, and was mainly 0.003-0.004 from October to December 2017, and thus still significantly above the stratospheric background (0.001-0.002). Stratospheric particle extinction coefficients (532 nm) were as high as 50-200 $Mm^{-1}$ until the beginning of September and of the order of 1 $Mm^{-1}$ (0.5-5 $Mm^{-1}$) from October 2017 until the end of January 2018. The corresponding layer mean particle mass concentration was of the order of 0.05-0.5 $\mu g\ cm^{-3}$ over months. Soot particle (light-absorbing carbonaceous particles) are efficient ice-nucleating particles (INPs) at upper tropospheric (cirrus) temperatures and available to influence cirrus formation when entering the tropopause from above. We estimated INP concentrations of 50-500 $L^{-1}$ until the first days in September and afterwards 5-50 $L^{-1}$ until the end of the year 2017 in the lower stratosphere for typical cirrus formation temperatures of $-55°C$ and an ice supersaturation level of 1.15. The measured profiles of the particle linear depolarization ratio indicated the predominance of non-spherical smoke particles. The 532 nm depolarization ratio decreased slowly with time in the main smoke layer from values of 0.15–0.25 (August-September) to values of 0.05–0.10 (October-November) and <0.05 (December-January). The decrease of the depolarization ratio is consistent with **aging of the smoke particles, growing of coating around the solid black carbon core (aggregates), and thus change of the shape towards a spherical form.** We found ascending aerosol layer features over the most southern European stations, especially over the Eastern Mediterranean at 32-35°N, that ascended from about 18-19 km to 22-23 km height from the beginning of October to the beginning of December 2017 (about 2 km per month). **Because of the similarity with observations of rising aerosol layer heights after major volcanic eruptions during the autumn and winter seasons when tropical aerosols are transported northward driven by the Brewer Dobson circulation (BDC), we concluded that the BDC was also responsible for the observed ascending stratospheric smoke layer structures. The important role of BDC to influence aerosol transport**

in the northern hemisphere is further corroborated by recent studies about the fire smoke plumes as discussed in this article.

## 1 Introduction

Record-breaking wildfire activity in British Columbia during the summer of 2017 coupled with rather favorable weather conditions on 12 August 2017 triggered the evolution of five major thunderstorms over western Canada in the afternoon of this day (Peterson et al., 2018). Exceptionally strong and well organized pyrocumulonimbus (pyroCb) clusters (Fromm et al., 2010; Peterson et al., 2017) developed over the fire areas and lifted enormous amounts of fire smoke into the upper troposphere and lower stratosphere (UTLS) (Khaykin et al., 2018; Peterson et al., 2018; Ansmann et al., 2018; Hu et al., 2019). Within pyroCbs the upward transport takes usually less than one hour from the ground to the tropopause level (Fromm et al., 2000, 2003; Rosenfeld et al., 2007). Many of the smoke particles may have served as cloud condensation nuclei (CCN) and ice-nucleating particles (INP), but the amount of smoke particles was so large that most of them were just lifted as interstitial aerosol to the UTLS region. The particles apparently reached the stratosphere as pure soot particles **(light-absorbing carbonaceous particles formed from incomplete combustion) (Petzold et al., 2013) and had a non-spherical shape after 7-10 days of travel towards Europe (Haarig et al., 2018; Hu et al., 2019) and even after months as will be shown here.** Self-lifting effects **caused by sunlight absorption and warming of the ambient air** (Boers et al., 2010; Siddaway and Petelina, 2011; de Laat et al., 2012) then led to a further significant ascent of the soot-containing layers. The aerosol optical thickness (AOT) in the UTLS region must have exceeded values of 2–3 at 500 nm wavelength so that strong absorption in the visible spectrum and warming of the smoke layers occurred and enabled the fire smoke plumes to ascend by about 2-3 km per day during the first days after injection as was observed with the spaceborne lidar aboard CALIPSO (Cloud-Aerosol Lidar and Infrared Pathfinder Satellite Observation) (Khaykin et al., 2018).

Peterson et al. (2018) and **Yu et al. (2019)** discussed the strength of this stratospheric smoke event based on spaceborne lidar observations and passive remote sensing and concluded that the pyroCb driven aerosol injection into the UTLS was comparably with a moderate volcanic eruption characterized by a Volcanic Explosivity Index of 3-4. The 12 August 2017 event, denoted as Pacific Northwest Event by Peterson et al. (2018), injected 0.1-0.3 Tg of total aerosol particle mass into the lower stratosphere. However, such mass comparisons do not provide an adequate description of the difference regarding the impact on atmospheric processes. Volcanic and smoke particles show rather different chemical, physical and morphological properties. In contrast to liquid, spherical, less light-absorbing sulfuric acid droplets of volcanic origin, stratospheric smoke particles from the wildfires in 2017 were observed to be non-spherical (Haarig et al., 2018; Hu et al., 2019), **and probably consisted of a solid core (black carbon (BC) aggregate) with non-spherical organic coating (Yu et al., 2019)**. In contrast to volcanic sulfuric acid particles, soot particles significantly absorb solar radiation **(direct effect on climate)**, and also influence the evolution of cirrus clouds by serving as INP in heterogeneous ice nucleation processes **(indirect effect)** (Hoose and Möhler, 2012; Kanji et al., 2017; Ullrich et al., 2017), in contrast to liquid sulfuric acid droplets which influence cirrus occurrence and evolution via homogeneous ice nucleation (Jensen and Toon, 1992; Sassen et al., 1995; Liu and Penner, 2002) and **(Campbell et al., 2012)**. **Homogeneous**

**nucleation is the process in which droplets freeze (and no solid particle phase is involved). In the case of heterogeneous nucleation, a solid particle is needed to initiate the nucleation of an ice crystal, but can take place at much lower ice supersaturation as needed to iniate homogeneous freezing.**

After injection on 12 August 2012, the smoke traveled to northern Canada, and then through the jet stream eastward, crossed the North Atlantic and reached Europe on 21-23 August 2017 (Ansmann et al., 2018; Haarig et al., 2018; Zuev et al., 2019; Hu et al., 2019). Compared to the maximum stratospheric perturbation over Europe after the eruption of Mt. Pinatubo in 1991 (Ansmann et al., 1997), 20 times higher particle extinction coefficients were observed in the lower stratosphere over Germany on 22 August 2017 (Ansmann et al., 2018). The smoke spread over the entire northern hemisphere during the following weeks, mostly at heights below 20 km with the dominating westerly air flow, and reached even the tropics via the dynamical transport around the Asian monsoon anticyclone (Kloss et al., 2019). The strong stratospheric perturbation diminished slowly from September 2017 to May-June 2018 according to SAGE III-ISS (Stratospheric Aerosol and Gas Experiment III mounted aboard the International Space Station) and OMPS-LP (Ozone Mapping Profiler Suite Limb Profiler onboard Suomi National Polar-orbiting Partnership, Suomi NPP) observations (Kloss et al., 2019; Yu et al., 2019). A fraction of the smoke particles ascended to heights of 20-23 km and enriched the natural soot particle reservoir located between 20 and 30 km height (Renard et al., 2008). The stratospheric smoke influenced radiative transfer (Ditas et al., 2018; Kloss et al., 2019; Yu et al., 2019), chemical processes (Yu et al., 2019), and probably cirrus evolution after entering the upper troposphere via gravitational settling and other processes causing an effective downward transport.

This historical event of a strong and long-lasting stratospheric aerosol perturbation was observed all over Europe with ground-based lidar systems of the European Aerosol Research Lidar Network (EARLINET) (Pappalardo et al., 2014). The arrival of first optically dense smoke plumes and layers over France was documented by Khaykin et al. (2018) and Hu et al. (2019) and over Central Europe in the accompanying articles of Ansmann et al. (2018) and Haarig et al. (2018). As a highlight, the smoke particles could be characterized regarding size distribution and shape properties at Leipzig (Germany) and Lille (France) by means of triple-wavelength polarization lidar observations (Haarig et al., 2018; Hu et al., 2019).

In this study, we present the observations from August 2017 to the end of January 2018 and discuss the decay phase and the changing optical and microphysical properties of the smoke particles over the almost six-month period. A strong role in the long-lasting 2017-2018 monitoring effort was played by the subnet PollyNET (NETwork of POrtabLe Lidar sYstems) (Baars et al., 2016) which consists of continuously running multiwavelength polarization/Raman lidars. The smoke layers were well detectable even six months after the injection until the end of January 2018. We will discuss the stratospheric perturbation in terms of layer base and top heights, AOT and column mass load, vertically mean extinction coefficients and soot mass concentrations, and even in terms of INP concentration estimates.

Note that the first lidar network observations of a pyroCb-related stratospheric smoke event was presented by Fromm et al. (2008) after the Chisholm Fire event in May 2001. Six lidar stations at Ny-Ålesund on Spitsbergen, at Esrange, Sweden, the EARLINET sites of Kühlunsgborn and Garmisch-Partenkirchen, Germany, at Boulder, Colorado, and Mauna Loa, Hawaii, observed the meridional spread of smoke from 20°N to 79°N. The dispersion over the 60° belt occurred within one month. Smoke features persisted on a hemispheric scale for at least 3 months. The German EARLINET station of Garmisch-Partenkirchen has

also one of the longest stratospheric aerosol records in the world (Trickl et al., 2013). The effects of major volcanic eruptions (El Chichon, Pinatubo) and numerous moderate volcanic eruptions are documented at Garmisch-Partenkirchen in southern Germany since January 1975. Also longer periods of quiescent times (to determine the minimum stratospheric aerosol background levels) are included in the almost 40-year data record.

## 2 Lidar networks: EARLINET and PollyNET

Twenty-three EARLINET stations from northern Norway (at 69°N) to Cyprus (34.5°N) (Pappalardo et al., 2014) contributed to the study. The lidar stations are shown in Fig. 1 together with five additional non-EARLINET lidar stations at Hatfield (UK), Lindenberg (Germany), El Arenosillo (Spain), Kosetice (Czech Republic), and Haifa (Israel). These non-EARLINET stations are closely collaborating with the EARLINET team under the umbrella of the European infrastructure project ACTRIS (Aerosols, Clouds, and Trace gases Research InfraStructure, https://www.actris.eu/) which is a pan-European initiative consolidating actions amongst European partners producing high-quality observations of aerosols, clouds and trace gases. ACTRIS is composed of observing stations, exploratory platforms, instrument calibration centers, and a data center, and aims to support atmospheric and environmental science by providing a platform for researchers to combine their efforts more effectively. Different lidar systems are operated at the EARLINET stations but the quality assurance / quality control (QA/QC) programs developed and run within ACTRIS RI (research infrastructure) allow a continuous control of the lidar operation and data. Among the considered 28 stations, there are seven continuously measuring Polly instruments (Althausen et al., 2009; Engelmann et al., 2016; Baars et al., 2016) operated at Kosetice, Limassol (Cyprus), Haifa, Warsaw (Poland), Hohenpeissenberg (Germany), Evora (Portugal) and at Finokalia on Crete (Greece).

Most of the EARLINET aerosol lidars are not designed for stratospheric aerosol observations. They are optimized for tropospheric measurements (boundary layer (BL) aerosols, diurnal cycle of BL conditions, lofted dust plumes in the free troposphere). The Polly instruments have, e.g., a 30 cm telescope and a small laser emitting light pulses of 110 mJ at 532 nm at a repetition rate of 20 Hz. In contrast, the big Leipzig EARLINET lidar (Mattis et al., 2010; Haarig et al., 2018; Jimenez et al., 2019) has an 80 cm telescope, 500 mJ per laser pulse at 532 nm and a repetition rate of 30 Hz. This lidar is highly capable to monitor stratospheric aerosol even at background conditions (Mattis et al., 2010; Finger, 2011). Most measurements presented in Sect. 3 are performed with the Polly instruments. Long averaging times (of typically 3-6 hours during nighttime hours) and vertical smoothing lengths of several 100 m had to be applied in the data analysis. An example is shown in Fig. 2. Fortunately, aerosol layering structures in the stratosphere are usually long-lasting, coherent, and persistent over many hours and sometimes even over days so that long temporal averaging and signal smoothing will not remove essential information about the observed stratospheric smoke layers.

In Sect. 3, quality-assured lidar observations are presented and discussed, mostly based on the retrieval of particle backscatter coefficients and particle linear depolarization ratio at 532 nm. Details of the lidar data analysis can be found in D'Amico et al. (2015, 2016); Mattis et al. (2016); Freudenthaler (2016); Baars et al. (2016); Mamouri and Ansmann (2016, 2017). The EARLINET observations are taken from the EARLINET data base (EARLINET, 2019) and selected by careful inspection

of the involved lidar teams. The Polly data analysis was performed by following the EARLINET data analysis protocols and procedures. **The Raman lidar method (Baars et al., 2016) was used to compute the particle backscatter coefficient from the ratio of elastic-backscatter to the respective nitrogen Raman signal.** To compute and correct for molecular backscatter and extinction contributions to the lidar backscatter signals, GDAS (Global Data Assimilation System) air temperature and

pressure data were partly used in the Polly data analysis (GDAS, 2019). However, most days were analyzed by assuming standard atmospheric conditions in the stratosphere. Significant differences to the results obtained with GDAS data were not observed. **The relative uncertainties are of the order of 5-10% in the case of the particle backscatter coefficient and AOT, and <5% for the particle depolarization ratio.**

Except the Lindenberg and Payerne (Switzerland) lidars, all participating stations provided height profiles of the particle

backscatter coefficient $\beta_p$ at 532 nm. Several stations could successfully measure the 532 nm volume (i.e., Rayleigh + particle) linear depolarization ratio and the respective particle depolarization ratio in the stratosphere. The powerful lidar system of the Meteorological Observatory Lindenberg of the German Weather Service provided the optical properties at 355 nm. The height profiles of the particle backscatter coefficient were used to determine base height $z_{bot}$ and top height $z_{top}$ of the detected stratospheric smoke layers. In the next step, the layer mean and column-integrated smoke optical properties were calculated.

The relative uncertainties in the lidar products shown in Sect. 3 are of the order of 10-30% (particle backscatter coefficients), 20-50% (particle extinction estimates and AOT estimates), and 10-30% for the presented smoke layer mean values for the particle depolarization ratio. The smoke layer geometrical properties (base and top heights) may have an uncertainty of the order of 50-150 m. The larger uncertainties describe the data quality for the observational period from October 2017 to January 2018. Signal noise is the main contributor to the large uncertainties.

Figure 2 presents an example of a complete Polly data analysis. A stratospheric smoke observation taken at Limassol, Cyprus, from 18:00–24:00 UTC on 9 September 2017 is shown. Height-time displays of the range-corrected signals indicated almost constant backscatter conditions over the six-hour period. The six-hour mean lidar return signals were vertically smoothed with a gliding averaging window length of 367.5 m before the calculation of the particle optical properties as a function of height above sea level (a.s.l.). The volume linear depolarization ratio at 532 nm in Fig. 2a, simply obtained from the calibrated cross-

to co-polarized signal ratio, enabled us to unambiguously identify the smoke layer in most cases. By means of the profiles of the volume depolarization ratio and the particle backscatter coefficient (in Fig. 2b) the particle linear depolarization ratio $\delta_p$ at 532 nm in Fig. 2c was calculated. **When the particle depolarization ratio exceeds a threshold value of e.g., 0.02, non-spherical particles are detected (Haarig et al., 2018). The depolarization ratio information is used to determine bottom and top height of each detected smoke layer.** The indicated base and top heights, $z_{bot}$ and $z_{top}$, of the smoke layer in Fig. 2c

are the mean values obtained from several 60-90-minute mean backscatter profiles measured from 18:00-24:00 UTC. **Smooth (instead of sharp) layer boundaries are the result of vertical signal smoothing with a window length of 367.5 m.**

Based on the profiles in Fig. 2, layer mean values of the particle backscatter coefficient $\overline{\beta_p}$ and particle linear depolarization ratio $\overline{\delta_p}$ were computed (as given in Fig. 2). By assuming an appropriate smoke extinction-to-backscatter ratio (lidar ratio) of 65 sr at 532 nm (Haarig et al., 2018), we obtained the aerosol optical thickness (AOT) $\tau_p$ and the layer mean particle extinction

coefficient $\overline{\sigma_p}$ also given in Figure 2. The relative uncertainties in the layer mean optical properties are of the order of 20-50%.

An overview of all lidar products together with the needed input parameter assumptions is given in Table 1. The listed input parameters were used throughout the investigated period from August 2017 to January 2018 and applied to all EARLINET data shown below.

By means of the computed optical properties, the microphysical properties, i.e., the soot mass concentration $M_\mathrm{p}$ and the ice-nucleating particle concentration $n_\mathrm{INP}$ were estimated. Here, conversion factors such as the soot particle extinction-to-volume conversion factor $c_\mathrm{v}$ and extinction-to-surface conversion factor $c_\mathrm{s}$ (Mamouri and Ansmann, 2016, 2017), the density $\rho_\mathrm{p}$ of the soot particles or the mass-specific extinction coefficient $k_\mathrm{ext}$ are required. From the measured smoke optical properties and the derived microphysical properties (from multiwavelength lidar inversions), presented by Haarig et al. (2018) for the optically dense smoke layer observed over Leipzig, Germany on 22 August 2017, the extinction-to-volume and extinction-to-surface conversion factors $c_\mathrm{v}$ and $c_\mathrm{s}$ in Table 1 were obtained. The soot particle density is highly variable and can vary from 0.2–2 $\mu$g cm$^{-3}$ (Rissler et al., 2013). As a compromise, we selected arbitrarily a value of 0.9 $\mu$g cm$^{-3}$ in our study. Similarly, the mass-specific extinction coefficient can vary from about 3 m$^2$ g$^{-1}$ to >15 m$^2$ g$^{-1}$ (Smith et al., 2015; Forestieri et al., 2018). Thus, the mass concentration estimation is highly uncertain (factor of 2-3). The INP concentration $n_\mathrm{INP}$ (see Table 1) is computed by using an INP parameterization developed for heterogeneous ice nucleation on soot particles via deposition nucleation (i.e., direct deposition of water vapor on the INPs) (Ullrich et al., 2017). Input aerosol parameter is the particle surface area $s_\mathrm{p}$. In addition, the atmospheric conditions (ambient temperature $T$ and ice supersaturation $S_\mathrm{ice}$ within the cirrus layer) are considered in the $n_\mathrm{INP}$ calculation via the $\eta_\mathrm{dep}$ term (see Table 1). The INP efficacy of aerosol particles increases by an order of magnitude when the temperature decreases by 5 K and is thus highest at tropopause level (coldest point of the troposphere). This behavior is described by the $\eta_\mathrm{dep}$ term (Mamouri and Ansmann, 2016; Ullrich et al., 2017). The relative uncertainty of the entire INP retrieval is determined by the large uncertainty (factor of 2-5) in the INP parameterization (Ullrich et al., 2017).

## 3 Observations

### 3.1 Decay of the stratospheric perturbation

Figure 3 provides an overview of the stratospheric smoke observations conducted with the 28 lidar systems. We considered all observations above 10 km height (a.s.l.) during the first four weeks after injection (until 9 September 2017). Afterwards (since 10 September), only the layers clearly above the local tropopause are shown. Vertical lines represent individual observations (one per day and site) of the detected smoke layers from base to top. The observations were taken after sunset and signal averaging time periods were at least 2 hours, with only a few exceptions. We subdivided the EARLINET observations according to the colors used in Fig. 1 for northern Europe (black, Norway), central and western Europe (green), the Iberian peninsula (blue, Spain and Portugal) and for southeastern Europe (red, mainly central and Eastern Mediterranean stations). Most of the Polly observations will be presented in Fig. 4 and are given here as grey background lines.

Smoke was frequently detected all over Europe until the end of October 2017 as the dense set of colored vertical lines indicates. Within a few weeks, the smoke spread over large parts of the northern hemisphere. This quick dispersion is corrob-

orated by the lidar observations aboard the CALIPSO satellite (Kar et al., 2019) in August and September 2017. **Based on atmospheric modelling and spaceborne extinction measurements (SAGE III-ISS), Yu et al. (2019) showed that the fire plumes reached the latitudes from 30-70°N within the first two weeks after the Pacific Northwest Event on 12 August 2017. A fast spread over the northern hemisphere** was also reported by Fromm et al. (2008) after the Chisholm pyroCB-related stratospheric smoke event in 2001. In northern Norway (69°N), the 2017 smoke layer was observed below 16 km height, whereas over the central, western and southern European stations (excluding here the Polly instruments), the smoke reached 22 km height. Also the spaceborne lidar shows this height dependence in terms of zonal averages of the attenuated total-to-Rayleigh backscatter ratio (Kar et al., 2019). According to the ground-based lidar observations in Fig. 3 the layer depth was frequently 1-2.5 km and in some cases even more than 5 km.

The Polly observations in Fig. 4a collected at Evora (Portugal), the central European stations of Hohenpeissenberg (Germany), Kosetice (Czech Republic), **and Warsaw (Poland)** and in the Eastern Mediterranean **(Finokalia on the Greek island of Crete, Limassol in Cyprus, and Haifa, Israel) also show that the layer top frequently exceeded 20 km (up to around 23 km) from mid September 2017 until the end of the year. Similarly, Yu et al. (2019) found the maximum top height at 23 km by using the spaceborne SAGE III-ISS aerosol extinction observations.** The main smoke layer extended from 15 and 20 km height. The smoke was frequently detected over southwestern and central Europe in the beginning of the smoke period (August-September 2017), and then mostly over the Eastern Mediterranean (October 2017 to January 2018). The data analysis was stopped at the end of January 2018 because no significant smoke layer was found anymore over Finokalia, Limassol, and Haifa during the following months. The results are again in agreement with the spaceborne lidar observations of the zonally averaged smoke optical properties and the detected latitudinal differences regarding occurrence, height, and vertical depth of the smoke layers in the months from September to November 2017 (Kar et al., 2019).

As shown in Fig. 4b, the stratospheric AOT at 532 nm decreased rapidly from values >0.2 in August 2017 to values between 0.005 and 0.03 in the beginning of September 2017, and afterwards the AOT ranged from 0.002 (almost stratospheric background conditions) to 0.008 with most values between 0.003 and 0.004 (over Finokalia, Limassol and Haifa, mid September to December 2017). A lidar ratio of 65 sr was applied to the respective column-integrated particle backscatter coefficients, integrated over the vertical column from $z_{bot}$ to $z_{top}$ (see Fig. 2), to obtain the AOT values. The AOT fluctuations are partly caused by the relatively strong impact of signal noise on the retrieval results. However, also atmospheric variability contributed to the observed fluctuations and to the respective vertically mean extinction coefficients (mean backscatter coefficient for the vertical column from $z_{bot}$ to $z_{top}$ multiplied with the soot lidar ratio of 65 sr) shown in Fig. 4c. We observed vertically mean 532 nm particle extinction coefficients for the smoke layers from 10-200 $Mm^{-1}$ in August 2017, 2-50 $Mm^{-1}$ until 5 September 2017, 1-10 $Mm^{-1}$ until the end of September, and finally values from 0.5-5 $Mm^{-1}$ (accumulating around 1 $Mm^{-1}$) until the end of January 2018.

532 nm AOT values around 0.004 indicate already typical stratospheric aerosol conditions for periods without major volcanic eruptions as discussed in Trickl et al. (2013) and in further articles reviewed and summarized in Ansmann et al. (2018). Based on 731 clear sky EARLINET nighttime lidar observations at Leipzig from January 2000 to June 2008 we conclude, however, that the minimum stratospheric AOT is of the order of 0.001 to 0.002 for the layer from 1 km above the tropopause to the

top of the identified aerosol structures (<30 km height) (Finger, 2011). This is in accordance with the long-term observations presented by Trickl et al. (2013) and spaceborne stratospheric background observations presented by Kloss et al. (2019) and Vernier et al. (2018). When using a typical extinction-to-backscatter ratio of 50 sr (for non-soot particles), the vertical mean particle extinction coefficients at minimum stratospheric aerosol conditions are in the range of 0.1–0.2 $Mm^{-1}$ at 532 nm (Finger, 2011).

**Fig. 5 provides a statistical overview of smoke layer depths. 566 daily Polly observations (conducted at the seven Polly stations after sunset from August 2017 to January 2018) of individual layers were analyzed. As shown, the vertical extent of the smoke layers was between 500 and 1500 m in 50% of all cases. However, smoke layer depths of several kilometers were observed as well.**

We compared our findings with measurements of the particle extinction coefficient at 521 nm wavelength aboard the International Space Station (ISS, SAGE III) presented by Yu et al. (2019); Kloss et al. (2019). For the more homogeneous period from October to December 2017, Kloss et al. (2019) found the main smoke layer also between 15 and 20 km height. They analyzed stratospheric extinction measurements for an area from 25–38°N and 40-95°E (covering the Middle East, Central Asia, and western China). The Polly stations at Cyprus and Israel were just west of this data analysis region. The mean extinction coefficients for this large area of almost 1000 km × 5000 km were about 0.5-1 $Mm^{-1}$ during the October-December period, and hence in good agreement with the Polly observations. The good agreement also indicates that the assumed smoke lidar ratio of 65 sr at 532 nm is justified. For the entire northern hemisphere (>40°N), Kloss et al. (2019) found mean particle extinction coefficients of 0.8-1 $Mm^{-1}$ for the October-December period and for the height range from about 14-19 km. According to the SAGE III and OMPS-LP observations stratospheric background extinction values were again reached in April-May 2018, about 8–9 months after the intense smoke injection event.

**Yu et al. (2019) presented mean smoke extinction coefficients (at 1020 nm) at 18 km height for the northern latitudes from 15-60°N as a function of time. From these observations we can conclude that the maximum 18 km mean extinction coefficient at 532 nm was close to 1 $Mm^{-1}$ (in October 2017), and accumulated around 0.5–0.7 $Mm^{-1}$ during the following months until the end of 2017. The stratospheric background (0.2-0.25 $Mm^{-1}$ at 532 nm after Yu et al. (2019)) was almost reached in May 2018.**

After conversion of the smoke extinction coefficients into respective mass concentrations (see Sect. 2 for more details) we found smoke mass concentrations of the order of 1–25$\mu$g $cm^{-3}$ in August and the beginning of September (see Fig. 4c), and afterwards frequently values from 0.1–1 $\mu$g $cm^{-3}$. Minimum stratospheric background values are <0.02 $\mu$g $cm^{-3}$ (Finger, 2011). Column mass concentrations exceeded 10 mg $m^{-2}$ in August 2017, and later on most values were found in the range from 0.1–1 mg $m^{-2}$ (see Fig. 4b). The particle mass estimates are uncertain by a factor of 2-3 due to the unknown soot particle density.

Figure 6 highlights the potential of soot particles to serve as INP and to discuss the potential impact on ice formation at tropopause level. The extinction coefficients in Fig. 4c were converted to INP concentrations for a typical cirrus formation temperature of $-55$°C and typical supersaturation conditions expressed by $S_{ice} = 1.15$. Besides slow downward motion by gravitational settling of the soot particles, an efficient way to transport aerosol from the lower stratosphere downward to the

upper troposphere are stratospheric intrusions (Trickl et al., 2014, 2016). Smoke particles reaching the upper troposphere and entrained into ascending humid tropospheric air masses may trigger cirrus formation at conditions with ice supersaturation values <1.4, still not favorable for homogeneous ice nucleation which needs ice supersaturation levels of typically 1.5-1.7. Heterogeneous ice formation on soot particles may thus have slightly enhanced cirrus formation in the northern hemisphere, especially during the first few months after injection.

The observed smoke extinction coefficients indicate INP concentrations of 3000 $L^{-1}$ in the beginning of the event during August 2017, then 50-500 $L^{-1}$ until 5 September, 10-300 $L^{-1}$ until 20 September, 5-50 $L^{-1}$ until November, and finally <20 $L^{-1}$ until the end of January 2018 for $T = -55°C$ and $S_{ice} = 1.15$. These values are large and can sensitively disturb cirrus formation in the usually clean upper troposphere.

## 3.2 Particle shape and size characteristics

**Haarig et al. (2018) and Hu et al. (2019) discussed the shape properties of the fire smoke particles based on lidar observations over Europe about two weeks after injection into the UTLS regime over Canada. They found high particle linear depolarization ratios (PLDR) at 355 nm (mostly 0.2-0.25) and 532 nm (0.15-0.2), and low values of 0.03-0.07 at 1064 nm for the smoke in the stratosphere. The high depolarization values at 355 and 532 nm indicate, first of all, that the particles were non-spherical. Ideal spheres such as liquid cloud droplets and wet marine particles would produce particle depolarization ratio close to zero. However, besides the strong sensitivity of PLDR to particle shape, also particle size influences PLDR (Mamouri and Ansmann, 2014, 2017). Fine-mode mineral dust particles (diameters $< 1~\mu m$) cause PLDR values around 0.15 at 532 nm whereas coarse-mode mode dust particles lead to PLDR of 0.35-0.4 according to laboratory studies and field observations as reviewed by (Mamouri and Ansmann, 2014, 2017). The dependence on size caused the observed strong wavelength dependence of PLDR of the stratospheric smoke plumes over Europe in August 2017 as pointed out by Haarig et al. (2018). The size distribution mainly consisted of a well developed accumulation mode. A coarse mode was absent. The inversion of the multiwavelength extinction and backscattering data revealed that the particles had diameters from 400 to 1400 nm with the mode maximum at 600-700 nm (Haarig et al., 2018). Particles with diameters $<1~\mu m$ thus dominated backscattering of the laser radiation. However, if coarse-mode particles dominate as in the case of typical desert dust size distributions, the PLDR wavelength dependence is less pronounced (Haarig et al., 2017).**

**Recently, Yu et al. (2019) modeled the optical properties of the Canadian smoke particles. They assumed that in the beginning an external mixture, consisting of (a) so-called fractal BC particles (i.e., fractal aggregates of BC) coated with organics causing an overall spheroidal shape and (b) organic particles without BC, rapidly coagulated and left behind mixed organic/BC particles with a typical abundance of 2% BC. The authors concluded from their modeling studies that the observed high PLDR of 0.2 at 532 nm (Haarig et al., 2018; Hu et al., 2019) cannot be explained by the presence of fractal BC particles with non-spherical organic coating alone, due to their small abundance. They hypothesize that the organic coated particles were most likely solids because they either froze in the stratosphere or effloresced.**

Gialitaki et al. (2019) modeled the optical properties (PLDR and lidar ratio at 355, 532, and 1064 nm) of the aged non-spherical smoke particles and compared the results with the respective multiwavelength lidar observation presented by Haarig et al. (2018). These extensive simulations for particle effective radii of 550 nm suggest that the smoke particles were compact and almost spherical in shape.

In Fig. 7, we now provide an overview of all available 532 nm depolarization ratios measured with the ACTRIS/EARLINET lidar consortium from August 2017 to the end of January 2018. A few 355 nm particle depolarization values are included (Lindenberg). Most values are contributed by the Polly lidars. In the beginning, orange and red colors prevailed. The retrieved particle depolarization ratios were between 0.15–0.25 at 532 nm. Because the tropospheric lidars were not optimized for stratospheric observations (at relatively low backscatter and AOT conditions), a significant contribution of signal noise to the variability in the depolarization ratio values has to be considered. However, a general trend, i.e., a decrease of the depolarization ratio with time towards 0.05–0.1 and later <0.05 is clearly visible. **This decrease of the layer mean depolarization ratio is probably mainly related to a growing coating of the smoke particles. The larger the coating shell is, the higher is the probability that the particles are perfectly spherical. However, also the removal of the larger smoke particles by sedimentation may have contributed to the decrease of PLDR. As explained above, the depolarization ratio decreases with decreasing contribution of large particles to light backscattering.**

### 3.3 Underlying transport processes

**In the following, we discuss a variety of processes that influence the aerosol transport and observed aerosol properties and features in the lower stratosphere. In Fig. 8, we show again the Polly observations of the smoke layer structures, but now only for the southern most stations in Portugal and the Eastern Mediterranean. At the southern European sites, coherent observations without strong disturbances by extended cloudy periods and unfavorable weather conditions in autumn and winter were possible. Such coherent measurements were not possible at the more northern stations, e.g., in Germany and Poland, so that many of the very thin aerosol features were not detectable at these more northern stations. As can be seen, many individual and apparently randomly distributed fire smoke layers are visible in Fig. 8. The prevailing westerly winds (jet stream) caused a main stratospheric aerosol transport from west to east. A descending trend (downward moving of the layer) as usually found after major volcanic eruptions as a result of sedimentation of particles (Jäger, 2005) was not visible from September 2017 to January 2018 in Fig. 8. Gravitational settling and warming of the air mass by absorption of solar radiation by the soot particles may have compensated each other. The decrease of the depolarization ratio over the months may thus be mainly related to the change of particle shape due to particle aging processes as suggested by Yu et al. (2019) and Kloss et al. (2019). However, during the autumn and winter season (from mid September to end of December) a northward transport of aerosols from the tropical stratospheric reservoir (TSR) towards the mid latitudes must be considered. Such a meridional aerosol transport out of the tropics was observed in any winter half year after major volcanic eruptions such as the El Chichón (in 1982) and Pinatubo eruptions (in 1991) and is related to the Brewer-Dobson circulation (Jäger, 2005).**

### 3.3.1 Brewer-Dobson circulation

The Brewer–Dobson circulation (BDC) describes the global-scale meridional circulation of the stratosphere (Seviour et al., 2012; Butchart, 2014; Abalos et al., 2015). BDC is characterized by tropospheric air rising into the stratosphere in the tropics, moving poleward before descending in the middle and high latitudes. As described by Jäger (2005), the merdional aerosol transport out of the TSR is modulated by the quasi-biennial oscillation (QBO) of the equatorial lower and middle stratosphere with alternating (and descending) regimes of easterly and westerly winds. The aerosol transport out of the TSR is suppressed when strong horizontal wind shear during the easterly phase of the QBO separates the tropics from the extra-tropical westerlies, while equatorial westerlies reduce the wind shear and promote transport into the winter hemisphere by isentropic mixing due to planetary waves penetrating into the subtropics and tropics and breaking there.

As pointed out by Jäger (2005), a very similar aerosol transport out of the tropics was observed over Garmisch-Partenkirchen (southern Germany, $47.5°N$) in the first winter after the major volcanic eruptions of El Chichón and during the second winter after the Pinatubo eruption caused by similar phases of the QBO with strong westerly winds at the relevant aerosol layer heights. During the second winter after the Pinatubo eruption, a clear and continuous rise of the aerosol layer top height by about 5 km from the beginning of October 1992 (25 km layer top) to the end of December 1992 (30 km layer top) was observed. QBO-related westerly winds also prevailed in the winter of 2017-2018 (https://acd-ext.gsfc.nasa.gov/Data_services/met/qbo/qbo.html), however only in the lower part of the QBO regime (from about 17 to 23 km height). At greater heights, strong easterly winds prevailed. So, the northward movement of tropical aerosols was favored up to 23 km and suppressed higher up. Thus, the rising layer height from 17-18 km in the beginning of October 2017 to 22–23 km at the beginning of December 2017 observed over the Eastern Mediterranean Polly stations and shown in Fig. 8 were probably caused by the QBO-influenced BDC.

As outlined by Kloss et al. (2019), the fire plumes injected into the lower stratosphere at high northern latitudes in August 2017 partly reached the tropics. The transport to the tropics was mediated by the anticyclonic flow of the Asian monsoon circulation. The fire plume reached the Asian monsoon area in late August/early September, when the Asian Monsoon Anticyclone (AMA) was still in place. A substantial part of the fire plume was entrained in the anticyclonic flow at the AMA edge and transported around its eastern flank into the tropics, where the air has further been lifted with the ascending branch of the BDC and then transported from the extra-tropics to the tropics. In the tropics, the fire plumes were lifted by about 5 km in seven months (September to April) in the upwelling branch of the BDC. Based on SAGE III-ISS extinction observations a slope in the aerosol signal with downwelling velocities (at northern latitudes $> 40°N$, 5 km in three months, October to January) and upwelling velocities (in the tropics, 0-25°N, September to April) was found. Thus, Kloss et al. (2019) hypothesize that the BDC played a sensitive role in both extra-tropical downward and tropical upward transport of the aerosol.

The SAGE III measurements showed that the main smoke layer (for an area over Asia, 5-25°N, 45-95°E), i.e., of 1000 km $\times$ 5000 km with center over India) ascended from approximately the tropopause at 16-17 km (10 September

2017) to about 20-21 km height (10 January 2018) and thus with an average speed of the order of 1000 m per month in the tropics. A steady lifting of the smoke layers in the tropics by 50-100 m per day (over several months) as part of BDC and the subsequent northward transport of the lifted layers can thus produce an apparently ascending smoke feature as observed over Limassol (34.5°N) and Haifa (32.5°N). The later in the year the northward transport starts towards the north, the longer was the time for lifting before the northward advection occurred.

However it remains an open question whether the coherent (apparently ascending) structures observed over Evora in September (blue in Fig. 8) and the apparently ascending features observed over the eastern Mediterranean (with a shift of about three weeks) belong together and are (as a whole) related to the beginning BDC winter circulation. Evora is about 4000 km west of Cyprus and Haifa. Difference in the planetary wave structures and phases, modulating even the meridional stratospheric air flow, may be the reason for the shift between the ascending aerosol features over eastern Atlantic (Evora) and the Eastern Mediterranean (Middle East region).

All in all, a coherent picture of the BDC impact on the stratospheric smoke transport was found with upward motion in the tropics (Kloss et al., 2019), and downward motion at northern and high northern latitudes (Kloss et al., 2019; Yu et al., 2019), and a horizontal transport to the north at sub-tropical midlatitudes of 32-35°N, i.e., without a descending trend as found further to the north.

However, many more rising, mostly short-term features are visible in Fig. 8, especially before the BDC became activated in autumn 2017. An example of an ascending short-term fire smoke structure was observed over Kosetice from 21–23 August 2017 and discussed by Ansmann et al. (2018). The layer was found at 12 km on 21 August 2017 and then coherently went up to 16 km within two days. This behavior could be explained by the fact that the wind velocity decreased with height from the tropopause (jet stream region) to 16 km height. Even if a compact smoke plume starts at all heights in the lower stratosphere simultaneously over the fire region, strong vertical wind shear may produce an apparently ascending aerosol layer several 1000 of kilometers downwind. Over Kosetice, the layers close to the tropopause traveled much faster because of the strong wind of 50 m s$^{-1}$ than the smoke layers at 15-16 km height where the horizontal wind component showed values around 15-20 m s$^{-1}$. Because of this strong vertical wind shear the smoke layer arrived over Kosetice one day later at 15-16 km height than at 12 km height (Ansmann et al., 2018). This vertical wind shear effect may explain ascending features observed over days, but cannot explain ascending features lasting over months.

### 3.3.2 Lifting by gravito-photophoresis forces

Two further smoke lifting processes are discussed in the literature. The first one is related to the gravito-photophoresis effect (Rohatschek, 1996; Pueschel et al., 2000; Cheremisin et al., 2005, 2011). Upward motion of individual particles is caused by radiometric forces resulting from normal stresses on the particle surface due to temperature gradients in the gas surrounding the surface. Gas molecules continuously impact on the surface of an aerosol particle and are reflected (Cheremisin et al., 2011). During reflection the molecules may pick up some energy and leave the surface with a higher thermal energy. The required temperature gradients are produced by a difference in the thermal accomodation coefficient (in case of particle lifting the

accomodation coefficient at the bottom of the particle is higher than at its top). The Sun is the source of irradiance and negative photophoresis takes place, that is, a force pointing to the sun poses a lifting component that opposes the forces of gravity (Rohatschek, 1996; Pueschel et al., 2000). Very specific aerosol and atmospheric conditions must be fulfilled. Only particles well-aligned in the air flow can be lifted. Stable alignment (and lifting) is only possible in the case of larger particles for 5   which the center of gravity is then always below their geometrical center (i.e., in the lower half of the particles during lifting). However, if particles are too large and thus too heavy, gravitational settling will always dominate. Optimum sizes (diameters) of particles for lifting are 1-2 $\mu$m. A stable equilibrium with the force of gravity pointing to the Earth and the photophoretic force pointing upward will build up for these particles. However, as shown by Pueschel et al. (2000) for irregularly shaped stratospheric soot particles (chains of spherules) with sizes or lengths of 1 $\mu$m the resulting vertical velocity is 0.009 cm/s or 10   about 7-8 m per day at heights around 20 km. Thus the gravito-photophoresis effect cannot explain the found strong upward movement of the order of 70-80 m per day.

### 3.3.3   Self-lifting by absorption of solar radiation

Another process leading to a cross adiabatic movement (by diabatic heating) is related to the so-called self-lifting effect (Boers et al., 2010; Siddaway and Petelina, 2011; de Laat et al., 2012). Absorption of shortwave solar radiation heats the smoke layers 15   and creates buoancy that can then result in an ascent of the aerosol layer over several kilometers altitude within 1-2 days (Siddaway and Petelina, 2011; de Laat et al., 2012). Such a heating is seasonally dependent. The largest lifting effect occurs in the summer hemisphere around 21 June when aerosol layers are exposed to sunshine for close to 24 hours a day. Boers et al. (2010) demonstrated in the case of soot (assuming a single scattering albedo of 0.75 at 500 nm) for mid summer conditions at 40°N (approximately for Evora, Finokalia, Limassol, and Haifa in late summer) that an ascent rate of 2.5 km per day is 20   possible in the case of a smoke AOT of 3.5 (at 500 nm). For an AOT of 0.5, lifting is of the order of 400-500 m per day, and for an AOT of 0.003-0.005, a lifting velocity of a few meters per day during mid summer conditions is plausible. However, the strong lifting over Finokalia, Limassol and Haifa was observed in autumn (from October to December). In conclusion, the self-lifting effect can also be ruled out as an explanation for the measured upward movement of smoke layers in October to December 2017.

### 25   4   Conclusions

The spread of extremely high amounts of wildfire smoke injected into the UTLS over western Canada in August 2017 and the decay of the stratospheric perturbation were monitored and documented with a network of 28 ground-based lidars in Europe. Stratospheric soot layers were observed for six months from August 2017 to January 2018. The AOT decreased from initial values of >0.2 (in the second half of August) to 0.005-0.03 in the beginning of September 2017 and then to around 30   0.003-0.004 during the following months until January 2018. Layer mean extinction coefficients and soot mass concentrations were of the order of 1 Mm$^{-1}$ and 0.1 $\mu$g m$^{-3}$ over the months, respectively, and thus significantly above the minimum stratospheric aerosol background values (0.1-0.2 Mm$^{-1}$, 0.01-0.02 $\mu$g m$^{-3}$). The decrease of the particle linear depolarization

ratio with time was found to be best consistent with aging of the smoke particles and related changes in the smoke particle shape properties (from non-spherical to spherical particle shape). The estimated ice-nucleating particle (INP) concentration levels were significantly enhanced for several months and thus the smoke plumes served as a long-lasting reservoir for INPs able to trigger heterogeneous ice nucleation and in this way to influence cirrus formation at tropopause level. **The BDC was**

**found to influence the stratospheric aerosol transport and aerosol conditions during the autumn and winter months in agreement with recently published studies on the smoke event.**

**It would be interesting to find indications for an impact of smoke particles on ice formations at tropopause level. The most favorable time period for such a study is probably the first month (mid August to mid September 2017) after the pyroCB event on 12 August 2017, when the smoke particle number and thus the INP concentration was high enough**

**over northwestern Canada and downstream towards Europe and Asia to significantly influence cirrus formation at tropopause level. However, there is a controversial discussion whether additional INP lead to a substantial change in cloud life time, and cirrus optical and radiative properties. According to the classical and well established cloud seeding hypothesis, heterogeneous ice formation usually leads to a reduction of cirrus life time (Storelvmo et al., 2018) because the number of nucleated ice crystals is much lower than in the case of homogeneous freezing. Consequently,**

**the lower number of crystals leads to faster crystal growth and removal by sedimentation so that the cirrus layer dissolves more quickly. However, based on lidar observation of thin cirrus formation in the volcanic ash after the Eyjafjallajökull volcanic eruption Seifert et al. (2011), and as recently also pointed out by Ansmann et al. (2019), it was found that cirrus lifetime may be prolonged by heterogeneous ice nucleation when an almost unlimited reservoir of INP is available and the necessary water vapor supply is given over long time periods. Virga were missing in the**

**volcanic-ash-influenced cirrus case study of Seifert et al. (2011) and indicated a quite large number of rather small ice crystals. On the other hand, it is expected that heterogeneous ice nucleation usually leads to a low ice crystal number concentration in combination with large crystal sizes caused by fast growth in ice supersaturated air. So it remains open how the additional soot INP in the upper troposphere influence cirrus formation and properties. Studies with passive remote sensing from space and active remote sensing from ground preferably with lidar and cloud radar during August**

**and early September 2017 would be very useful to clarify this aspect.**

This record-breaking stratospheric smoke event is the second major event after the Eyjafjallajökull volcaninc eruption in 2010 (Ansmann et al., 2011; Pappalardo et al., 2013; Navas-Guzmán et al., 2013) that highlights the importance, need, and usefulness of EARLINET, a well-organized, Europe-wide, ground-based aerosol profiling network of advanced lidars. Dense sets of height and temporally resolved measurements of geometrical, optical, and microphysical smoke particle properties

were collected to document this event of a significant stratospheric perturbation in the northern hemisphere, to support aerosol transport and life cycle modeling with global atmospheric circulation models (Earth System Models, covering aerosol long-range transport, spread, and removal, and the influence of the aerosol layers on climate-relevant processes), and also to support spaceborne remote sensing of aerosols by providing high quality ground-truth data. The PollyNET observations have shown that automated, continuously running lidar systems are powerful tools and allow us to cover the decay phase of the stratospheric

aerosol perturbation in a coherent way. Without having continuous measurements, the smoke layering details and properties

as presented and discussed in this article would widely remain undetected. At European level, an upgrade of the current lidar capabilities is foreseen in terms of aerosol observation in the implementation of ACTRIS as research infrastructure. In this framework it is aimed to move towards powerful and continuously running automated lidars.

The research on this spectacular case of a stratospheric perturbation is ongoing and will be widely based on spaceborne active and passive remote sensing in combination with ground-based remote sensing (EARLINET and further ground-based lidar observations, e.g., in Asia and North America). This will then provide a good basis for sophisticated aerosol modeling. The complex transport features and climatic influences of stratospheric soot layers make it necessary to compare simulated smoke scenarios and the evolution of the smoke layer during long-range transport with the available observations. In this context one should finally mention (as an outlook what is left to be improved) that the realization of a well-organized ground-based lidar network such as EARLINET, but on a hemispheric or even global scale (as the Global Aerosol Watch (GAW) initiative GALION: GAW Aerosol LIdar Observations Network) (Bösenberg et al., 2008) would be desirable and could be seen as a big step forward towards a complete monitoring of global aerosol distributions and environmental conditions in the troposphere and stratosphere around the world. Sawamura et al. (2012) demonstrated the importance of having such a coordinated lidar profiling effort in the case of the Nabro volcanic eruption event. The importance for the need of such global aerosol monitoring network structures may increase during the upcoming years because of the hypothesis that in a changing climate natural hazards such as severe wildfires combined with pyroCb activity and major desert dust outbreaks may occur more frequently and that detailed profile observations are required to support weather and climate research and forecast. **Regarding vertically resolved observational studies of atmospheric processes (aerosol and cloud life cycles, aerosol-cloud-dynamics relationships) there is practically no alternative to ground-based (lidar and radar) network observations. Spaceborne lidars such as the CALIPSO lidar are complementary to these network observations by providing global 3-D aerosol distributions, but these snapshot-like satellite lidar observations are of limited use in process studies.**

Future activities should also be undertaken in the direction of harmonization of lidar network observations and data. In this sense, the effort to develop standardized tools for aerosol lidar analysis, as realized in the case of ACTRIS/EARLINET in form of the single-calculus-chain (SCC) software (D'Amico et al., 2015, 2016; Mattis et al., 2016), and to open its use to non-EARLINET lidar stations and teams is another step forward on the long way of global lidar data harmonization.

## 5 Data availability

EARLINET data are accessible through the ACTRIS data portal http://actris.nilu.no/. The long-term Polly lidar level-0 data are plotted online at polly.tropos.de, raw data are available at TROPOS upon request (polly@tropos.de). GDAS meteorological data can be downloaded at https://www.ready.noaa.gov/READYamet.php. GDAS1 data is available via the ARL webpage https://www.ready.noaa.gov/gdas1.php. All the analysis products are available at TROPOS upon request (info@tropos.de).

## 6 Author contributions

H.B. coordinated the project, communicated with all EARLINET groups and collected all EARLINET data. K.O. performed the Polly data analysis and prepared all figures supervised by H.B. and A.A. The layout of the manuscript was designed by A.A. and H.B. Finally, A.A. wrote the text in cooperation with H.B. and K.O. All EARLINET and further PollyNET co-authors performed the stratospheric smoke measurements, carefully analysed their observations with focus on stratospheric aerosol signatures and transfered the findings to TROPOS.

## 7 Competing interests

The authors declare that they have no conflict of interest.

*Acknowledgements.* The authors acknowledge support through ACTRIS under grant agreement no. 262 254 and BACCHUS (no. 603445) of the European Union Seventh Framework Programme (FP7/2007-2013), ACTRIS-2 and EXCELSIOR under grant agreement no. 654109 and 763643 from the European Union's Horizon 2020 research and innovation programme, and HD(CP)2 under grant agreement no. 01LK1502I by the German Ministry for Education and Research (BMBF). The research leading to these results has received funding also from the H2020 program of the European Union regarding GRASP-ACE (grant agreement no. 778349) and Spanish fundings (ref. TEC2015-63832-P, CGL2017-90884-REDT and MDM-2016-0600). Cork lidar observations were supported by the Irish Research Council (2014) and Science Foundation Ireland (05/RF/EEB011). The Cyprus observations were supported by the SIROCCO project which is co-funded by the Republic of Cyprus and the Structural funds of the European Union for Cyprus under the Research & Innovation Foundation grant agreement EXCELLENCE/1216/0217. Haifa lidar observations were funded by the German-Israeli Foundation (GIF grant I-1262 401.10/2014) with support of the Norman and Helen Asher Fund and Ollendorff Minerva Center. The development of the lidar inversion algorithm used to analyze Lille and Leipzig lidar data was supported by the Russian Science Foundation (project 16-17-10241).

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

**Table 1.** Lidar-derived smoke (soot) particle optical and microphysical properties and retrieval input parameters for 532 nm. $z_{\mathrm{bot}}$ and $z_{\mathrm{top}}$ are the layer base and top height of the detected stratospheric smoke layer, respectively. The values for the lidar ratio $S_{\mathrm{p}}$ and the two conversion factors $c_{\mathrm{v}}$ and $c_{\mathrm{s}}$ are taken from Haarig et al. (2018) and the estimate of particle density density $\rho_{\mathrm{p}}$ is based on Rissler et al. (2013). For detailed explanations see text.

| Parameter | Symbol | Value |
|---|---|---|
| Backscatter coefficient | $\beta_{\mathrm{p}}$ | |
| Integrated backscatter coefficient | $B_{\Pi,\mathrm{p}} = \int_{z_{\mathrm{bot}}}^{z_{\mathrm{top}}} \beta_{\mathrm{p}} dz$ | |
| Extinction coefficient | $\sigma_{\mathrm{p}} = S_{\mathrm{p}}\beta_{\mathrm{p}}$ | |
| Lidar ratio | $S_{\mathrm{p}}$ | 65 sr |
| Aerosol optical thickness (AOT) | $\tau_{\mathrm{p}} = \int_{z_{\mathrm{bot}}}^{z_{\mathrm{top}}} \sigma_{\mathrm{p}} dz$ | |
| Mass concentration | $M_{\mathrm{p}} = \rho_{\mathrm{p}} c_{\mathrm{v}} \sigma_{\mathrm{p}} = \sigma_{\mathrm{p}}/k_{\mathrm{ext}}$ | |
| Particle density | $\rho_{\mathrm{p}}$ | 0.9 g cm$^{-3}$ |
| Extinction-to-volume conversion factor | $c_{\mathrm{v}}$ | $0.1244 \times 10^{-12}$ Mm m$^3$ m$^{-3}$ |
| Mass-specific extinction coefficient | $k_{\mathrm{ext}} = 1/(\rho_{\mathrm{p}} c_{\mathrm{v}})$ | 8.93 m$^2$ g$^{-1}$ |
| Surface area concentration | $s_{\mathrm{p}} = c_{\mathrm{s}} \sigma_{\mathrm{p}}$ | |
| Extinction-to-surface conversion factor | $c_{\mathrm{s}}$ | $1.166 \times 10^{-12}$ Mm m$^2$ cm$^{-3}$ |
| Ice-nucleating particle concentration | $n_{\mathrm{INP}} = s_{\mathrm{p}} \eta_{\mathrm{dep}}(T, S_{\mathrm{ice}})$ | |

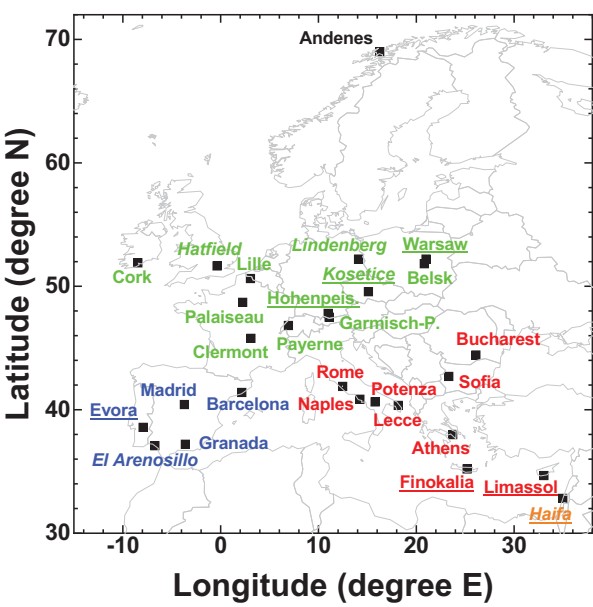

**Figure 1.** Lidar network of 23 ACTRIS/EARLINET stations and 5 non-EARLINET sites (in italics). This network observed stratospheric smoke layers from August 2017 to January 2018. Northern (black), central and western (green), southwestern (blue), southeastern European (red), and Israel (orange) lidar sites are distinguished. Polly stations are underlined.

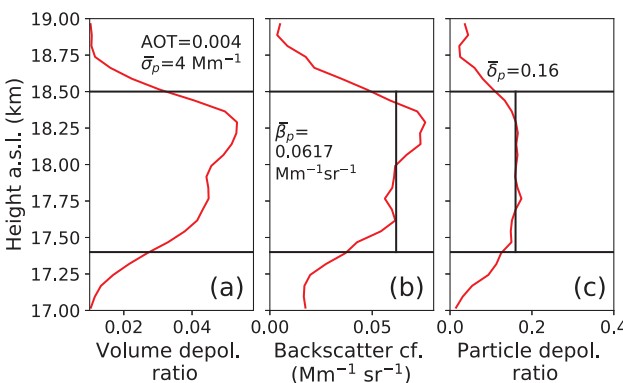

**Figure 2.** Analysis of a Polly measurement at Limassol, Cyprus, on 9 September 2017. The 532 nm backscatter and depolarization ratio profiles are computed from six-hour mean lidar return signal profiles (18:00-24:00 UTC). Vertical signal smoothing with a window length of 367.5 m is applied. The volume depolarization ratio in (a) and the particle backscatter coefficient in (b) were used to identify the smoke layer. The shown smoke layer base and top heights (horizontal lines) are mean values for the observation period, estimated from subsequent 60-90-minute mean depolarization ratio profiles. The particle depolarization ratio in (c) is the one for smoke (after the correction for Rayleigh depolarization contributions). Values for the vertically averaged particle extinction coefficient $\overline{\sigma_p}$ (for the column from $z_{bot}$ and $z_{top}$, assuming a lidar ratio of 65 sr) and 532 nm AOT, mean backscatter coefficient $\overline{\beta_p}$, and mean particle linear depolarization ratio $\overline{\delta_p}$ are stated in the three panels, respectively.

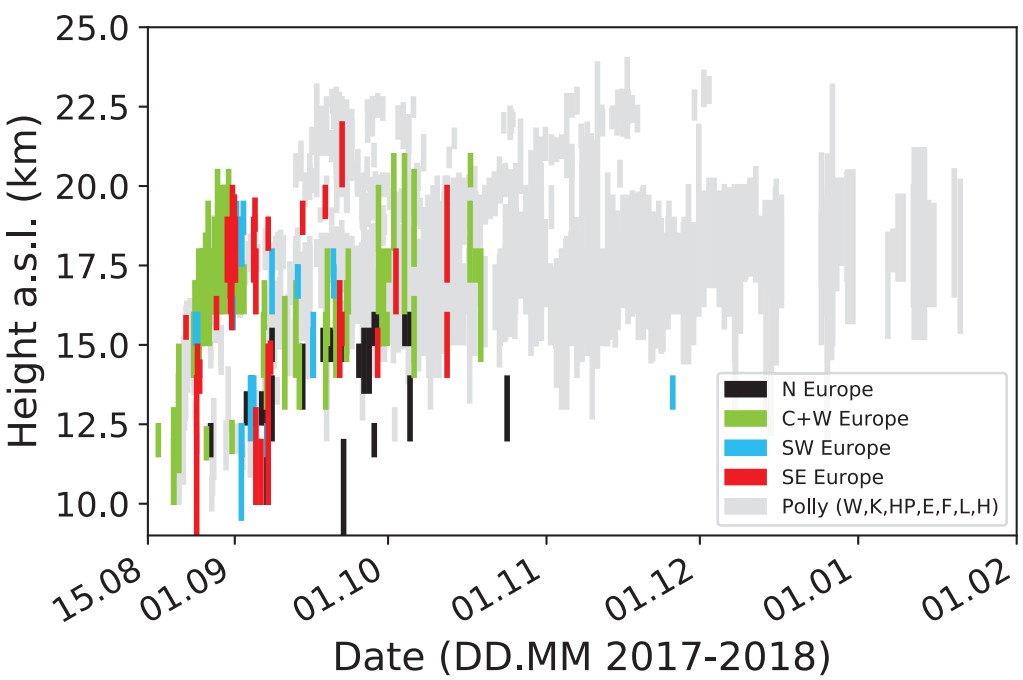

**Figure 3.** Overview of the lidar network observations of stratospheric smoke from August 2017 to January 2018. Each observation is considered as one colored vertical line indicating the vertical extent from layer base to top (in height above sea level, a.s.l.). One observation per day and site is considered. The colors separate the different European regions of the EARLINET stations as defined in Fig. 1. Polly observations (collected at Evora, Hohenpeissenberg, Kosetice, Warsaw, Finokalia, Limassol, and Haifa) are given here as grey background and are presented in Fig. 4.

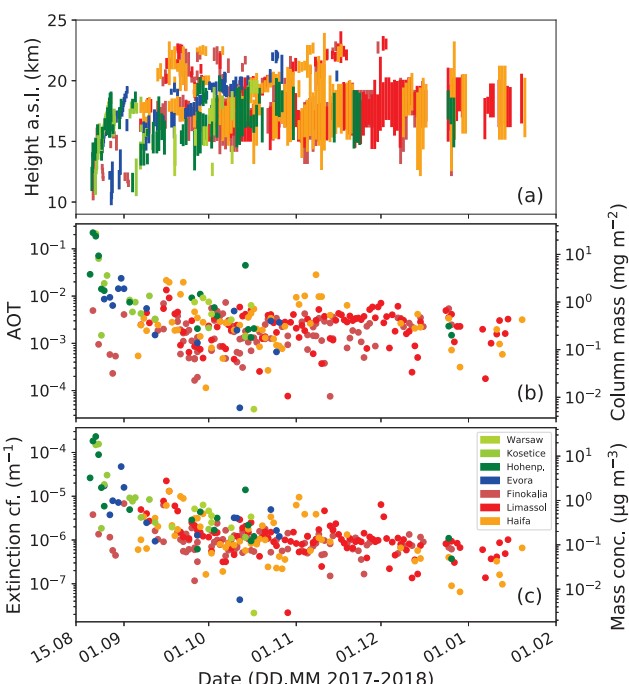

**Figure 4.** (a) Overview of all Polly observations of the stratospheric smoke layer (from base to top as colored vertical lines). For each station, one nighttime observation per day is considered. (b) Corresponding smoke layer AOT at 532 nm and estimated column-integrated smoke particle mass concentration, and (c) vertically averaged smoke particle extinction coefficient and corresponding mean particle mass concentration.

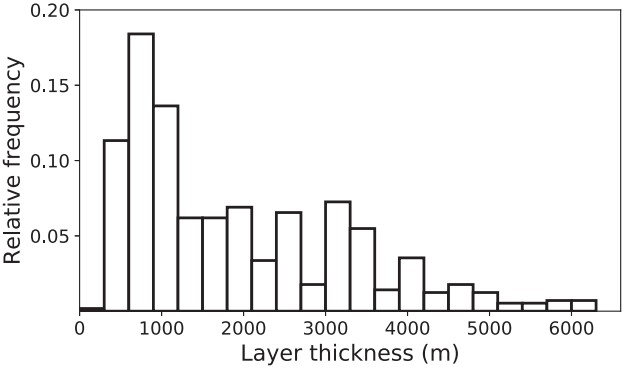

**Figure 5.** Frequency of occurrence of day-by-day smoke layer depth considering all 566 detected layers, based on all Polly observations at the seven sites from August 2017 to January 2018.

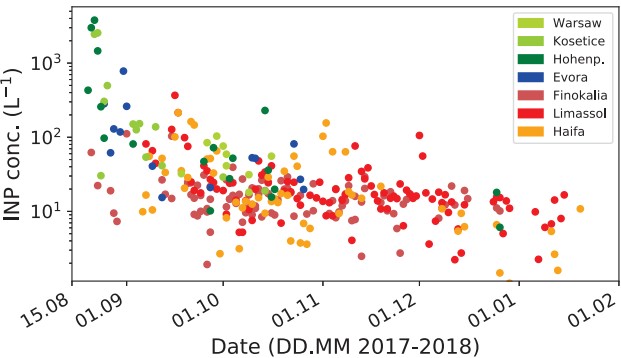

**Figure 6.** Ice-nucleating particle (INP) concentration estimated from the smoke extinction coefficients in Fig. 4(c), assuming heterogeneous ice nucleation (deposition nucleation) on soot particles at the temperature $T = -55°C$ and a typical ice supersaturation level during cirrus formation of $S_{\mathrm{ice}}$=1.15 (Ullrich et al., 2017).

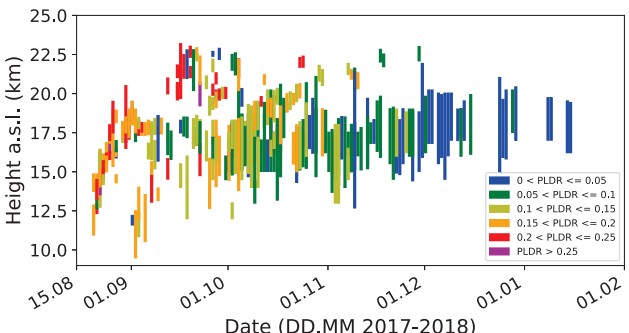

**Figure 7.** All individual day-by-day smoke observations of the 532 nm particle linear depolarization ratio (from all contributing stations, including several Lindenberg observations at 355 nm). Colors indicate different depolarization value ranges. The depolarization ratio decreased with time because of the removal of the larger non-spherical smoke particles and/or the change in the shape characteristics (from non-spherical to spherical particle shape).

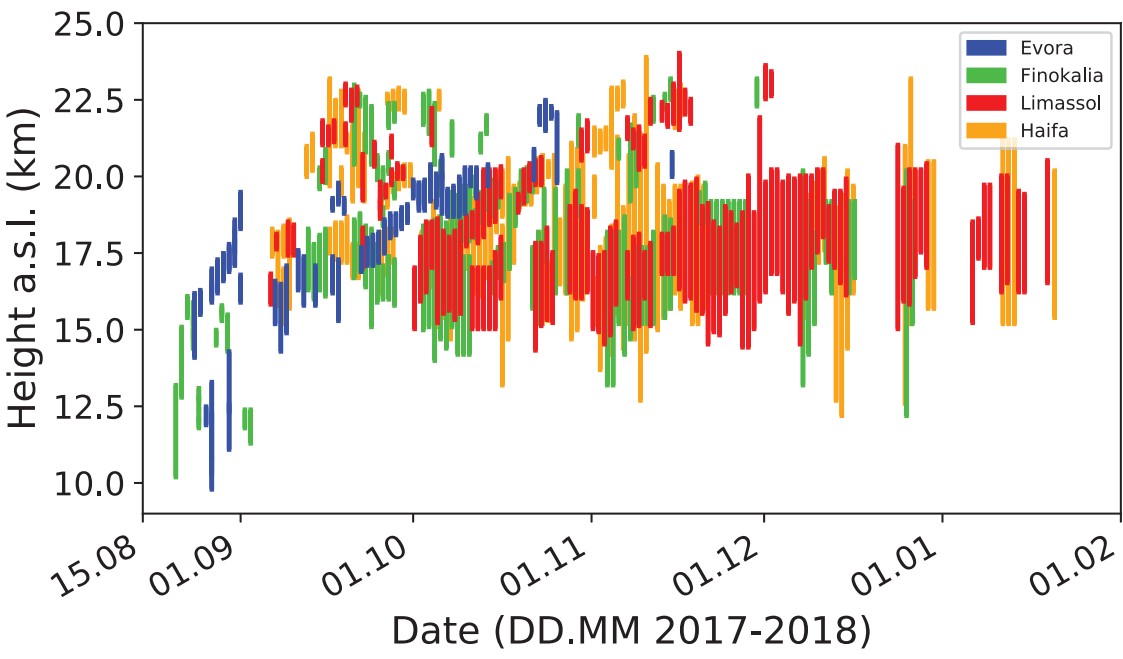

**Figure 8. Same as Fig. 4a, except for the southern Polly stations only. The Finokalia data set is shown here in green for better identification.The different data sets are shifted against each other by 6 hours and the linewidth is reduced to better see all observations. An ascending structure, first seen over Evora in September (in blue) and then also detected over the Eastern Mediterranean in October and November (in green, red, and ornage) triggered the discussion about potential aerosol lifting processes and effects.**