# Peer review of "The unprecedented 2017–2018 stratospheric smoke event: Decay phase and aerosol properties observed with EARLINET"

_Atmospheric Chemistry and Physics, 2019_

## Referee Comment (RC1) · Michael Fromm (Referee) · 12 Aug 2019

**Review of "The unprecedented 2017–2018 stratospheric smoke event: Decay phase and aerosol properties observed with EARLINET" by Baars et al. (hereafter B19).**

B19 offer a new contribution to the quite interesting set of studies examining a pyrocumulonimbus (pyroCb) smoke plume in the stratosphere that in several regards seems to have been unprecedented in remote sensing data sets going back decades. This work offers a valuable new perspective, involving two ground-based lidar networks: EARLINET and POLLYNET. There are additional additional lidars in Europe that are not involved in this work, but these two networks together have an impressive geographic spread over the continent and operation frequency that enables a sophisticated temporal and regional analysis of an evolving plume. Moreover, the capabilities of certain instruments (e.g. multispectral probing and polarization) give B19 the ability to assess microphysical and cloud-nucleation properties in addition to optical properties of an aerosol plume. An example of the strength of these networks as applied to this smoke event has already been demonstrated by several papers that B19 cite, wherein the nascent phase of this plume over Europe was characterized. Here they follow the plume until it was undetectable above noise levels, six months after the pyroCb injection.

The manuscript is well crafted, logically organized, straightforward to follow, and careful to characterize the levels of uncertainty in the data and processing methods.

B19 present valuable new results. This perspective of the decay of the Pacific Northwest Event (PNE) smoke plume illustrates new constraints on the sensitivity of these lidars to the stratosphere relaxing back to background conditions. They make a strong argument in the Conclusions section for the strategic value of these lidar networks, bolstered by the results they show. Hence this is appropriate for consideration in ACP. However, I have one major concern that keeps me from recommending this for publication. That will be discussed next, followed by a few minor and technical issues.

Major Concern:

Considerable attention is devoted to the visualization of a backscattering object referred to variously as, e.g. an "apparently ascending smoke layer," and "coherent, apparently upward moving structure." B19 then explore the most plausible physical mechanism for the supposed ascent. It has already been shown convincingly that the PNE smoke plume underwent diabatic ascent, in a paper cited by B19 (Khaykin et al. 2018). Khaykin et al. utilized a global aerosol data set (CALIPSO), which in my opinion is a requirement for ascertaining diabatic rise. Applying geographically bounded data sets such as these Eurocentric lidars to the task of quantifying diabatic ascent and assigning causal mechanisms is vulnerable to misinterpretation. Inferring diabatic rise from an upward sloping aerosol feature in an altitude vs. time analysis (using single or multiple lidars) is hampered by the additional plausible explanations for that slope that are impossible to resolve locally. For example a

sloping feature might simply be attributable to wind-speed shear operating on an evolving plume. For a plume below the jet max, the effect is an apparently descending slope. For a plume above the jet max, i.e. in the lower stratosphere, the effect would be an apparently ascending layer. The tropospheric example was brought up in the discussion of a previously published ACP paper:

https://www.atmos-chem-phys.net/10/11921/2010/acp-10-11921-2010-discussion.html

The stratospheric analog is shown in the following schematic.

**Plume above jet max (dv/dz < 0)**

[Figure]

[Figure]

Here is how the tilted plume would appear w.r.t. observations at a point location or limited geograqphic region.

Lidar Backscatter Time Series

Z

Time

The aerosol feature appears to be ascending but it's not.

B19 refer to an "ascending…layer" in the context of a time series plot comprising a season's worth of data. In the course of that time it is hard to conceive of the aerosols blowing over that limited domain as simply a "layer." The reader might take "layer" to mean a coherent physical object that is wholly within the sampling beam of those lidars. This of course is not what was occurring within the narrow string of lidars spanning the Mediterranean Sea and Iberia that B19 grouped for this season-long analysis. Over the course of the latter half of 2017 the smoke that blew into that regional swath had a vertical and horizontal history that is totally outside B19's domain.

Complicating the interpretation of diabatic ascent are two established facts about the PNE plume. 1. Above southern Europe the plume was as high as 20 km by late August, according to B19's Figure 3 and Khaykin et al. (2018), Figure . 2. The plume was already higher than 20 km in early September (Khaykin et al., 2018), when B19's nascent smoke signal is constrained to 15-17 km (Figure 7a--the figure central to B19's ascent argument). Hence it can be argued that the representation of smoke over the Mediterranean swath was biased quite low at the outset of the Fig. 7a time series. Given the results from southern France reported by Khaykin et al., B19 are compelled to argue for a physically-based distinction between the plume observed early over Iberia and eastern Mediterranean versus the plume in the French Mediterranean area in late August reported by Khaykin et al..

The fact that PNE smoke was higher than ~21 km in the extratropics by the first week of September means that any smoke observed at such altitudes at any time after that, in the extratropical latitudes of EARLINET and POLLYNET, may owe their altitude to a mechanism other than that put forward by B19. Wind shear meteorology may be one possible factor. The wind speed profile is also a climatological reality and hence may play a role in differential transport with respect to altitude on a longer time scale like that of Figure 7a. I'm not sure how much merit this has; I'd just ask B19 to consider broadening the discussion of the possible forces involved in an upwardly sloping feature such as that seen in Figure 7a.

B19 give a thorough survey of the mechanisms that might explain diabatic transport of absorbing aerosol such as smoke.  They find that the only candidate consistent with their data is a pathway proposed by Kloss et al. (2019) (K19). K19, which is at this time still under review, argue for ascent of the PNE smoke by way of a combination of horizontal transport to the tropics and subsequent diabatic ascent driven by the Brewer Dobson Circulation (BDC). If this is indeed the precursor condition and setup for B19's "apparent ascending layer," the entire upward slope in Figure 7a must be the consequence of a coherent, continuous flux of tropical air to B19's Mediterranean lidar belt during the entre fall and early winter of 2017. If this was the driving transport mechanism, it would also be necessary to argue why the "ascending layer" in Figure 7a had apparently little impact on any other European lidars near the Mediterranean belt and presumably downwind of that flux from the tropics. In short, B19's argument was not convincing to me.

I ran an experiment by computing back trajectories from an observation at El Arenosillo (NASA MPLNET) of smoke at 22 km on 25 October 2017, about midway along the Figure 7a time series. The purpose was to ascertain the general trajectory direction at that time/altitude. The results suggest that transport to the observation location was on westerly winds and did not appear to be consistent with flow from the tropics. Hence the smoke at that time over southern Iberia was more likely to have an extratropical history (at least for the preceding two weeks) than tropical. What is not shown be straightforward to ascertain is that along this path one will find CALIPSO aerosol-layer coincidences, at the approximate altitude of the 25 October layer, at several points along the trajectory path. By this example one can argue that individual observations in Figure 7a can be fully and quantitatively explained by simple meteorological transport, in this case from extratropical plume sightings. Separating this history from one imposed by the BDC would seem to be a tall order, and perhaps unnecessary. I'd ask B19 to comment on whether this experiment is well conceived, and if it raises a question as to the pathway of the smoke that is in the apparently ascending layer in Figure 7a.

[Figure]

The robustness of K19's analysis and conclusions is still in question, given that the manuscript is currently under review. Issues were raised with K19's analysis that are analogous to those raised in this review. The discussion is available here:

https://www.atmos-chem-phys-discuss.net/acp-2019-204/

To the extent that B19's conclusions w.r.t. plume ascent depend on the work of K19, it is suggested that B19 assess the issues raised in both reviews and then revisit the question of the causes of the upward slope in Figure 7a.

**Minor Concerns:**

Regarding the discussion of smoke ice-nucleating properties, B19 make a plausible hypothesis that the huge abundance of smoke at the tropopause could have had a discernible impact on cirrus formation over Europe (and presumably beyond). I would understand if B19 consider further exploration of this as beyond the scope of this paper. However it made me wonder if cirrus occurrence was indeed perturbed on fall 2017. Perhaps B19 could add a brief statement as to whether this is being explored or just generally a topic for future work.

P10, L20-22. Discussion of the relation of depolarization ratio to particle size. This is presented as if it is common knowledge, but I don't think it is. For instance, is this true for dust and volcanic ash? Please elaborate and/or cite the literature establishing this.

**Technical Concerns:**

**Abstract, L5 (and elsewhere): The term "soot" should be defined; otherwise it is ambiguous. E.g. sometimes "soot" is applied to aircraft emissions.**

**P2, L10: Change "ascent" to "ascend."**

**P2, L17: Change "2018" to "2017."**

**P3, L10: "**The particles obviously reached the stratosphere as pure soot particles…" What makes this "obvious?" Might it be better to use "apparently" instead?

P3, L12: Change "lead" to "led."

P3, L28: There is another paper that directly deals with this issue. Please consider citing Campbell et al. (2012). https://www.sciencedirect.com/science/article/pii/S135223101100968X

P4, L4: "part of the smoke particles" suggests a micro-level. Please reword.
**P5, L1: Is there a difference between "ACTRIS-2" and "ACTRIS" is used thereafter? If so, please clarify.**

**P5, L12: "PBL" should be spelled out at first usage.**

**P5, L26: "**analysis were performed" should be "analysis was performed"

P7, L18: "indicate" should be "indicates" to agree with the singular subject "set."

 P7, L21: "(because of the low tropopause height)" What does the tropopause height have to do with the altitude of the smoke layer? Please elaborate or reword.

**P11, L6: Change "ascend" to "ascent."**

**P12, L14: "…**could be lifted before." This is an incomplete sentence. Please modify.

**P12, L24: "The  unprecedented event of ..." is awkward. Perhaps "The unprecedented occurrence of…" instead?**

**References:** Gialitaki et al. Is it proper to cite a paper as "to be submitted"?

---

## Referee Comment (RC2) · Anonymous Referee #3 · 13 Sep 2019

The unprecedented 2017-18 stratospheric smoke event: Decay phase and aerosol properties observed with EARLINET

Holger Baars et al

This paper summarises measurements from the European lidar network of the notable event in 2017 when pyrocumulus convection over British Columbia lifted a thick smoke layer into the stratosphere. The paper concentrates on the decay phase of the evant and is mainly concerned with lidar measurements from the Polly network, which measured backscatter and depolarisation at 532 nm. Despite the fact that the Polly lidar is not optimised for stratospheric measurements, the smoke distribution in the period covered by the paper was sufficiently uniform that long exposure times could be used, and the paper provides a valuable summary of the distribution and some of the microphysical properties of the smoke as measured over Europe. The team is very experienced and for the most part I have only minor comments.

However, I have one major comment. The final two paragraphs of section 3.2, and all of section 3.3, are based on a particular interpretation of fig 4a – i.e. that the observed smoke layer consisted of a 'background' distribution of constant altitude, and a much thinner ascending layer, going from ~16 km in early September to 22 km in November. The paper claims that the depolarisation ratio is slightly higher for this layer, but fig. 6 does not really support this interpretation, especially when likely errors in depolarisation measurement above 20 km are considered. Otherwise there is little to distinguish the layers, apart from the eye of faith applied to Fig 4a. One could as easily say that a higher layer of particles appeared in November, or indeed that on occasion the particles were found above 20 km (mid-Sept, mid-Nov, early and late Dec). In summary, I find the evidence for a **coherent** ascending layer measured throughout this period to be very weak.

It might be appropriate to say that one interpretation of fig 4a is that there is an ascending layer, but to devote an entire section to the possible causes of the ascent is pushing the data far too much. A section on underlying transport processes is appropriate, but should give an overall discussion of the spread of the layer and its link to the dynamics, rather than the material on p.11. For example, why did the smoke layer linger a lot longer over the Eastern Mediterranean than over the Western Mediterranean? The authors should also be careful of terminology. The classical Brewer-Dobson circulation consists of ascent in the tropics to the mid-stratosphere, followed by poleward transport by the planetary-wave-driven meridional circulation, and descent at high latitudes. The transport being discussed here is in the lower stratosphere, mostly accomplished by synoptic-scale waves. The process described on p.12 is correct but it isn't the B-D circulation. Section 3.3 therefore requires a rewrite.

Minor comments

p.3. l.3 'then led to'

p.3. l.25 homogeneous nucleation is the process whereby droplets freeze without an external nucleus (i.e. a random nucleation process). Anything involving an external nucleus is heterogeneous

p.4 l.16 'effort was played by'

p.4 l.28 'stratospheric'

p.5 l.28 Lidar retrievals are generally sensitive to the assumed molecular density profile so the phrase 'significant differences' needs more quantitative elaboration

p.7 l.21 Why does the height of the tropopause affect the maximum height of the smoke layer? To first order, transport in the lower stratosphere is isentropic, so it is the tilt of isentropic surfaces that must be considered, not the tropopause (which crosses the isentropes).

p.7 l.22 'smoke reached 22 km'

p.7 l.25-28 What is the point of the Sicard reference? It adds nothing to the argument

p.7 l.30 'from' not 'since'

p.9 l.22 'introduction'

p.22 l.1 omit 'shown

p.22 l.6 'vertically average'

---

## Author Comment (AC1) · 26 Oct 2019

Dear Editor, Dear Reviewers!

Thank you for taking the time to carefully read the paper and provide us with many good comments and suggestions.

Before we present an item-by-item response, let us first summarize the main changes:

(1) We took the opportunity to additionally analyze the Polly data sets from Warsaw (Poland) and Finokalia (Crete, Greece) in detail. Now we have 7 Polly stations instead of 5 Polly sites. Accordingly, we changed the Figs. 3, 4, 5, 6, and 7 (in the revised version) and included the Warsaw and Finokalia observations into the shown Polly data sets.

(2) We skipped Fig. 7 (ascending layer) and Fig. 8 (main layer) of the discussion paper and therefore also Figs. 9b and 9c. Fig. 9a of the discussion version is now the new Fig. 5 (and includes the Warsaw and Finokalia layer depth observations).

(3) We provide more details on the underlying transport processes in Sect. 3.3 and better separate the different aspects and processes by introducing three subsections, 3.3.1: Brewer-Dobson circulation, 3.3.2: Lifting by gravito-photophoresis forces, 3.3.3. Self-lifting by absorption of solar radiation.

(4) We integrate the smoke observations and simulations of the optical properties (especially of the depolarization ratio and thus the impact of the smoke particle shape) presented in the Science paper of Yu et al. (2019) into the discussions and comparisons.

(5) The discussion of the impact of the Brewer Dobson circulation (BDC) is now intensified based on the fundamental JGR paper of Jaeger (2005). He showed in a very convincing way that BDC (in combination with the quasi biannual oscillation in the tropics, QBO) plays an important role in the meridional transport of aerosols out of the tropical stratospheric reservoir in every autumn and winter season. It is impossible to ignore the BDC effects and to neglect a discussion on BDC (and QBO) in our paper. And the ACP paper of Kloss et al. (2019) and the Science paper of Yu et al. (2019) clearly support our argumentation that the BDC had a strong impact on the aerosol transport from the tropics to high latitudes during the winter half year 2017-2018.

Now the step-by-step responses are listed. **Our answers are given in bold:**

**Reviewer #1 (M. Fromm):**

Baars et al. (B19) offer a new contribution to the quite interesting set of studies examining a pyrocumulonimbus (pyroCb) smoke plume in the stratosphere that in several regards seems to have been unprecedented in remote sensing data sets going back decades. This work offers a valuable new perspective, involving two ground-based lidar networks: EARLINET and POLLYNET. There are additional lidars in Europe that are not involved in this work, but these two networks together have an impressive geographic spread over the continent and operation frequency that enables a sophisticated temporal and regional analysis of an evolving plume. Moreover, the capabilities of certain instruments (e.g. multispectral probing and polarization) give B19 the ability to assess microphysical and cloud-nucleation properties in addition to optical properties of an aerosol plume. An example of the strength of these networks as applied to this smoke event has already been demonstrated by several papers that B19 cite, wherein the nascent phase of this plume over Europe was characterized. Here they follow the plume until it was undetectable above noise levels, six months after the pyroCb injection.

The manuscript is well crafted, logically organized, straightforward to follow, and careful to characterize the levels of uncertainty in the data and processing methods.

B19 present valuable new results. This perspective of the decay of the Pacific Northwest Event (PNE) smoke plume illustrates new constraints on the sensitivity of these lidars to the stratosphere relaxing back to background conditions. They make a strong argument in the Conclusions section for the strategic value of these lidar networks, bolstered by the results they show. Hence this is appropriate for

consideration in ACP.

**Thank you for these kind words!**

However, I have one major concern that keeps me from recommending this for publication. That will be discussed next, followed by a few minor and technical issues.

Major Concern:

Considerable attention is devoted to the visualization of a backscattering object referred to variously as, e.g. an "apparently ascending smoke layer," and "coherent, apparently upward moving structure." B19 then explore the most plausible physical mechanism for the supposed ascent. It has already been shown convincingly that the PNE smoke plume underwent diabatic ascent, in a paper cited by B19 (Khaykin et al. 2018). Khaykin et al. utilized a global aerosol data set (CALIPSO), which in my opinion is a requirement for ascertaining diabatic rise. Applying geographically bounded data sets such as these Eurocentric lidars to the task of quantifying diabatic ascent and assigning causal mechanisms is vulnerable to misinterpretation. Inferring diabatic rise from an upward sloping aerosol feature in an altitude vs. time analysis (using single or multiple lidars) is hampered by the additional plausible explanations for that slope that are impossible to resolve locally. For example a sloping feature might simply be attributable to wind-speed shear operating on an evolving plume. For a plume below the jet max, the effect is an apparently descending slope. For a plume above the jet max, i.e. in the lower stratosphere, the effect would be an apparently ascending layer. The tropospheric example was brought up in the discussion of a previously published ACP paper: https://www.atmos-chem-phys.net/10/11921/2010/acp-10-11921-2010-discussion.html

B19 refer to an "ascending…layer" in the context of a time series plot comprising a season's worth of data. In the course of that time it is hard to conceive of the aerosols blowing over that limited domain as simply a "layer." The reader might take "layer" to mean a coherent physical object that is wholly within the sampling beam of those lidars. This of course is not what was occurring within the narrow string of lidars spanning the Mediterranean Sea and Iberia that B19 grouped for this season-long analysis. Over the course of the latter half of 2017 the smoke that blew into that regional swath had a vertical and horizontal history that is totally outside B19's domain.

Complicating the interpretation of diabatic ascent are two established facts about the PNE plume. 1. Above southern Europe the plume was as high as 20 km by late August, according to B19's Figure 3 and Khaykin et al. (2018), Figure . 2. The plume was already higher than 20 km in early September (Khaykin et al., 2018), when B19's nascent smoke signal is constrained to 15-17 km (Figure 7a--the figure central to B19's ascent argument). Hence it can be argued that the representation of smoke over the Mediterranean swath was biased quite low at the outset of the Fig. 7a time series. Given the results from southern France reported by Khaykin et al., B19 are compelled to argue for a physically-based distinction between the plume observed early over Iberia and eastern Mediterranean versus the plume in the French Mediterranean area in late August reported by Khaykin et al..

The fact that PNE smoke was higher than ~21 km in the extratropics by the first week of September means that any smoke observed at such altitudes at any time after that, in the extratropical latitudes of EARLINET and POLLYNET, may owe their altitude to a mechanism other than that put forward by B19. Wind shear meteorology may be one possible factor. The wind speed profile is also a climatological reality and hence may play a role in differential transport with respect to altitude on a longer time scale like that of Figure 7a. I'm not sure how much merit this has; I'd just ask B19 to consider broadening the discussion of the possible forces involved in an upwardly sloping feature such as that seen in Figure 7a.

B19 give a thorough survey of the mechanisms that might explain diabatic transport of absorbing aerosol such as smoke. They find that the only candidate consistent with their data is a pathway proposed by Kloss et al. (2019) (K19). K19, which is at this time still under review, argue for ascent of the PNE smoke by way of a combination of horizontal transport to the tropics and subsequent diabatic ascent driven by the Brewer Dobson Circulation (BDC). If this is indeed the precursor condition and setup for B19's "apparent ascending layer," the entire upward slope in Fig. 7a must be the consequence of a coherent, continuous

flux of tropical air to B19's Mediterranean lidar belt during the entre fall and early winter of 2017. If this was the driving transport mechanism, it would also be necessary to argue why the "ascending layer" in Figure 7a had apparently little impact on any other European lidars near the Mediterranean belt and presumably downwind of that flux from the tropics. In short, B19's argument was not convincing to me.

I ran an experiment by computing back trajectories from an observation at El Arenosillo (NASA MPLNET) of smoke at 22 km on 25 October 2017, about midway along the Figure 7a time series. The purpose was to ascertain the general trajectory direction at that time/altitude. The results suggest that transport to the observation location was on westerly winds and did not appear to be consistent with flow from the tropics. Hence the smoke at that time over southern Iberia was more likely to have an extratropical history (at least for the preceding two weeks) than tropical. What is not shown be straightforward to ascertain is that along this path one will find CALIPSO aerosol-layer coincidences, at the approximate altitude of the 25 October layer, at several points along the trajectory path. By this example one can argue that individual observations in Figure 7a can be fully and quantitatively explained by simple meteorological transport, in this case from extratropical plume sightings. Separating this history from one imposed by the BDC would seem to be a tall order, and perhaps unnecessary. I'd ask B19 to comment on whether this experiment is well conceived, and if it raises a question as to the pathway of the smoke that is in the apparently ascending layer in Figure 7a.

The robustness of K19's analysis and conclusions is still in question, given that the manuscript is currently under review. Issues were raised with K19's analysis that are analogous to those raised in this review. The discussion is available here: https://www.atmos-chem-phys-discuss.net/acp-2019-204/

To the extent that B19's conclusions w.r.t. plume ascent depend on the work of K19, it is suggested that B19 assess the issues raised in both reviews and then revisit the question of the causes of the upward slope in Figure 7a.

**Thank you for your long explanations! Here is our summarizing answer and the details to the changes we made in the paper:**

**We removed Fig. 7 (ascending layer) and Fig. 8 (main layer) as strongly recommended by both reviewers. We agree with both reviewers to have a more careful and save argumentation, and to avoid a speculative discussion as much as possible. Now, we show a new figure (Fig. 7 in the revised version). This figure contains only Polly data from Evora, Portugal (Eastern Atlantic region) and the Eastern Mediterranean stations. Only the layering information is shown. The figure just shows the layering facts as they were observed! Although the coherent, ascending layering structures are clearly visible (to our opinion), we leave it more open to the reader to develop his/her opinion about the aerosol layering structures we observed.**

**In the revised manuscript, we simplified our discussion on underlying transport processes (section 3.3), but also intensified the discussion on the influence of the Brewer-Dobson circulation (BDC). To have a better structure of this long part of the paper, we introduce three subsections to better separate the different effects, processes and potential lifting aspects.**

**To our opinion, there is no way around discussing the Brewer-Dobson circulation (BDC) and its impact on the meridional aerosol transport out of the tropics to the mid latitudes. We cannot ignore the BDC (modulated by the quasi-biannual oscillation, QBO) and thus need to discuss this. The apparently ascending structures which were observed were most probably triggered by BDC-QBO effects.**

**Why is that the case?**

**We base our discussion on the fundamental study of Jaeger (JGR, 2005). Horst Jaeger investigated stratospheric aerosols over many decades (1980-2010) and convincingly explained how BDC, modulated by QBO, influences the aerosol transport out of the tropics. All his arguments are based on observations after major volcanic eruptions. We therefore provide the point of view of Jaeger (2005) now in Sect. 3.3.1. Our findings are in agreement with his argumentation. For example, the QBO in 2017-2018 favored a meridional (northward) aerosol transport below 23 km height and prohibited the**

**transport out of the tropics at heights above 23 km. Our observed smoke aerosol structures (ascending with time) were visible exactly up to 23 km. Furthermore, the observations of Kloss et al. (ACP, 2019, in press) show lifting of aerosol in the tropics and descending structures in the higher northern latitudes. Descending smoke structures were also found by Yu et al. (Science, 2019) at northern mid latitudes. All this is now mentioned in Sect. 3.3.1. These findings perfectly fit to this BDC influence. Kloss et al. (2019) explicitly mentioned the link to BDC in their paper. The Kloss et al. paper was recently accepted and will be published during the next 3-4 weeks.**

**To continue, we see a clear similarity between the ascending aerosol structures (top height) as found by Jaeger (2005) after the Pinatubo eruption (i.e., an increase of the Pinatubo layer top height by 5km from the beginning of October to the beginning of January) and our ascending smoke layer structures (of about 4-5 km from the beginning of October to the beginning of December in the Eastern Mediterranean). We checked the QBO and found the same westerly winds in autumn 2017 as after the Pinatubo event (in the second winter after the eruption, 1993, the most import one to discuss the QBO-BDC influence as Jaeger (2005) mentioned). Furthermore we found in the 2017-2018 QBO data that the easterly winds started above 23km in autumn 2017. As mentioned above, the easterly winds prohibit northward transport, whereas westerly winds favor the release of tropical aerosol towards the north. This is written in the paper of Jaeger (2005) and we repeat these statements in Sect. 3.3.1. Surprisingly, all smoke structures go up to 23 km, found by EARLINET and also found by Yu et al. (2019). So in conclusion, there are clear and strong indications that BCD effects are responsible for the aerosol structures we observed. And as mentioned, all this is in agreement with the findings and discussions presented by Kloss et al. (ACP, 2019, in press).**

**The mentioned vertical wind shear effect that YOU propose can only explain the observed ascending structures occurring over several days and shortly after the injection to the stratosphere. We observed such an effect on 21-23 August 2017 over central Europe and discussed that in Ansmann et al. (ACP, 2018). But, for a longer time period in which the aerosol has traveled several times around the globe, wind shear (strong winds close to the tropopause, and weak winds at 20-23 km, and present for months) cannot explain coherent and ascending smoke aerosol structures over weeks and months. We discuss this now briefly at the end of Sect. 3.3.1.**

Minor Concerns:

Regarding the discussion of smoke ice-nucleating properties, B19 make a plausible hypothesis that the huge abundance of smoke at the tropopause could have had a discernible impact on cirrus formation over Europe (and presumably beyond). I would understand if B19 consider further exploration of this as beyond the scope of this paper. However it made me wonder if cirrus occurrence was indeed perturbed on fall 2017. Perhaps B19 could add a brief statement as to whether this is being explored or just generally a topic for future work.

**We discuss this point in the conclusion section. There is no simple answer regarding the consequences of the impact of smoke on cirrus life time and properties. Therefore we can just start the discussion on a potential influence. We state now in the conclusions:**

**It would be interesting to find indications for the impact of smoke particles on ice formations at tropopause level. The most favorable time period for such a study is probably the first month (mid August to mid September 2017) after the pyroCB event on 12~August 2017, when the smoke particle number and thus the INP concentration was high enough over northwestern Canada and downstream towards Europe and Asia to significantly influence cirrus formation at tropopause level. However, there is a controversial discussion whether additional INP lead to a substantial change in cloud life time, and cirrus optical and radiative properties. According to the classical and well established cloud seeding hypothesis, heterogeneous ice formation usually leads to a reduction of cirrus life time (Storelvmo 2018) because the number of nucleated ice crystals is much lower than in the case of homogeneous freezing. Consequently, the low number of crystals leads to a faster crystal growth and removal by sedimentation so that the cirrus layer dissolves quickly. However, based on lidar observation of thin cirrus formation in the volcanic ash after the Eyjafjalljökull volcanic eruption (Seifert et al., 2011) and recently also pointed out by Ansmann et al. (2019), it was found that the cirrus lifetime may be**

**prolonged by heterogeneous ice nucleation when an almost unlimited reservoir of INP is available and the necessary water vapor supply is given over long time periods. Virga were missing in the volcanic-ash-influenced cirrus case study of Seifert et al. (2011) and indicated a quite large number of rather small ice crystals. On the other hand, it is expected that heterogeneous ice nucleation usually leads to a low ice crystal number concentration in combination with large crystal sizes caused by fast growth in ice supersaturated air. So it remains open how the additional soot INP in the upper troposphere influence cirrus formation and properties. Studies with passive remote sensing from space and active remote sensing from ground preferably with lidar and cloud radar during August and early September 2017 would be very useful to clarify this aspect.**

P10, L20-22. Discussion of the relation of depolarization ratio to particle size. This is presented as if it is common knowledge, but I don't think it is. For instance, is this true for dust and volcanic ash? Please elaborate and/or cite the literature establishing this.

**We extend the discussion accordingly and provide lab and field results and corresponding references to the dependence of depolarization ratio on particle size. Furthermore, we improved the discussion based on simulation studies of the smoke optical properties done by Yu et al. (2019) and Gialitaki, et al. (2019, now we take as reference of the ILRC conference proceedings, more details below).**

Technical Concerns:

Abstract, L5 (and elsewhere): The term "soot" should be defined; otherwise it is ambiguous. E.g. sometimes "soot" is applied to aircraft emissions.

**We give a definition for soot now (first paragraph in the Introduction section) based on the discussion and definitions in the paper of Petzold et al. (ACP, 2013).**

**Petzold, A., et al.: Recommendations for reporting "black carbon" measurements, Atmos. Chem. Phys., 13, 8365-8379, https://doi.org/10.5194/acp-13-8365-2013, 2013.**

P2, L10: Change "ascent" to "ascend."

**Done!**

P2, L17: Change "2018" to "2017."

**Done!**

P3, L10: "The particles obviously reached the stratosphere as pure soot particles…" What makes this "obvious?" Might it be better to use "apparently" instead?

**Agree!**

P3, L12: Change "lead" to "led."

**Done!**

P3, L28: There is another paper that directly deals with this issue. Please consider citing Campbell et al. (2012). https://www.sciencedirect.com/science/article/pii/S135223101100968X

**Included!**

P4, L4: "part of the smoke particles" suggests a micro-level. Please reword.

**Done!**

P5, L1: Is there a difference between "ACTRIS-2" and "ACTRIS" is used thereafter? If so, please

clarify.

**We avoid to write ACTRIS-2 now**.

P5, L12: "PBL" should be spelled out at first usage.

**Done!**

P5, L26: "analysis were performed" should be "analysis was performed"

**Done!**

P7, L18: "indicate" should be "indicates" to agree with the singular subject "set."

**Yes, changed!**

P7, L21: "(because of the low tropopause height)" What does the tropopause height have to do with the altitude of the smoke layer? Please elaborate or reword.

**Agree! We removed the tropopause statement.**

P11, L6: Change "ascend" to "ascent."

**Done!**

P12, L14: "…could be lifted before." This is an incomplete sentence. Please modify.

**Improved!**

P12, L24: "The  unprecedented event of ..." is awkward. Perhaps "The unprecedented occurrence of…" instead?

**We changed the text!**

References: Gialitaki et al. Is it proper to cite a paper as "to be submitted"?

**We take the 4-page contribution of Gialitaki  et al. (2019) now as reference. This contribution was presented at the ILRC in Hefei, China, in June 2019.**

**Reviewer #2:**

This paper summarises measurements from the European lidar network of the notable event in 2017 when pyrocumulus convection over British Columbia lifted a thick smoke layer into the stratosphere. The paper concentrates on the decay phase of the event and is mainly concerned with lidar measurements from the Polly network, which measured backscatter and depolarisation at 532 nm. Despite the fact that the Polly lidar is not optimised for stratospheric measurements, the smoke distribution in the period covered by the paper was sufficiently uniform that long exposure times could be used, and the paper provides a valuable summary of the distribution and some of the microphysical properties of the smoke as measured over Europe. The team is very experienced and for the most part I have only minor comments.

**Thanks for these kind words!**

However, I have one major comment. The final two paragraphs of section 3.2, and all of section 3.3, are based on a particular interpretation of fig 4a – i.e. that the observed smoke layer consisted of a 'background' distribution of constant altitude, and a much thinner ascending layer, going from ~16 km in early September to 22 km in November. The paper claims that the depolarisation ratio is slightly higher

for this layer, but fig. 6 does not really support this interpretation, especially when likely errors in depolarisation measurement above 20 km are considered. Otherwise there is little to distinguish the layers, apart from the eye of faith applied to Fig 4a. One could as easily say that a higher layer of particles appeared in November, or indeed that on occasion the particles were found above 20 km (mid-Sept, mid- Nov, early and late Dec). In summary, I find the evidence for a **coherent** ascending layer measured throughout this period to be very weak.

**You are right (as M. Fromm, reviewer #1)! Therefore we removed Fig. 7 (ascending layer) and Fig. 8 (main layer), but introduce a new figure (Fig. 7 in the revised version), showing the observations at Evora (Eastern Atlantic) and in the Eastern Mediterranean (Finokalia, Limassol, Haifa) only. We leave it in a more objective way open to the reader to see the ascending structures or not.**

**We continue with our reply after the next paragraph ….**

It might be appropriate to say that one interpretation of fig 4a is that there is an ascending layer, but to devote an entire section to the possible causes of the ascent is pushing the data far too much. A section on underlying transport processes is appropriate, but should give an overall discussion of the spread of the layer and its link to the dynamics, rather than the material on p.11. For example, why did the smoke layer linger a lot longer over the Eastern Mediterranean than over the Western Mediterranean? The authors should also be careful of terminology. The classical Brewer-Dobson circulation consists of ascent in the tropics to the mid-stratosphere, followed by poleward transport by the planetary-wave-driven meridional circulation, and descent at high latitudes. The transport being discussed here is in the lower stratosphere, mostly accomplished by synoptic-scale waves. The process described on p.12 is correct but it isn't the B-D circulation. Section 3.3 therefore requires a rewrite.

**Please see the discussion on the Brewer-Dobson circulation above (reply to reviewer #1 in this reply letter). The paper of Jaeger (JGR, 2005) clearly shows that BDC, modulated by QBO, is active in every autumn and winter period of each year. We discuss this in detail in Sect. 3.3.1.**

**As already mentioned above, but repeat it here: We provide a better structure of section 3.3. (underlaying transport processes). We introduce three subsections to have a better separation of the different effects and impacts.**

**Nevertheless, we still consider that the Brewer-Dobson circulation (BDC, modulated by the quasi-biannual oscillation, QBO) needs to be discussed as a potential smoke transport mechanism. Our argumentation is now based on the paper of Jaeger (2005) in Sect. 3.3.1. We think our argumentation is convincing now. We see a clear similarity between the ascending aerosol structures (top height of the Pinatubo aerosol layer) as found by Jaeger (2005) after the Pinatubo volcano eruption and our ascending smoke structures. Both ascended by about 4-5 km within the autumn and early winter period. We checked the QBO for 2017-2018 and found the same westerly winds in autumn 2017 as after the Pintubo event (for the relevant second winter, 1993, same QBO winds) but also that the easterly winds started above 23km in autumn 2017. Easterly winds prohibit northward transport. Thus, the BDC could have an influence up to 23 km only. And exactly to that height the ascending structures are visible in autumn 2017 over Evora and over Limassol/Haifa. So in conclusion, there are clear and strong indications that BCD effects are visible in the data. This is also mentioned in the study of Kloss et al. (ACP, 2019, in press), and of Yu et al., Science (2019). We state that in the Sect. 3.3.1.**

**While do we see differences (regarding the ascending structures) between Evora (eastern Atlantic) and Eastern Mediterranean? Here we have to say: We do not know! A potential reason may be that the position and phase of planetary waves, modulating the aerosol meridional transport, were quite different over the eastern Atlantik (close to Evora) and over the Eastern Mediteranean. Evora is about 4000 km west of Haifa. We state all this in Sect. 3.3.1.**

Minor

comments p.3.

l.3 'then led to'

**Improved!**

p.3. l.25 homogeneous nucleation is the process whereby droplets freeze without an external nucleus (i.e. a random nucleation process). Anything involving an external nucleus is heterogeneous

**Thank you for your explanation! We tried to improve the text. In model study publications (e.g., of Kaercher et al. and Jensen et al.) it is stated that homogeneous freezing is given when you have just ONE phase (thus even when sulfate particles freeze they call that homogeneous freezing). And only when a second phase comes into play (solid INP) then they call it heterogeneous ice nucleation.**

p.4 l.16 'effort was played by'

**Improved!**

p.4 l.28 'stratospheric'

**Improved!**

p.5 l.28 Lidar retrievals are generally sensitive to the assumed molecular density profile so the phrase 'significant differences' needs more quantitative elaboration

**We now provide numbers : about 10% for backscatter and AOT, and less than 5% in the case of the depolarization ratio. We analyzed about ten cases in different seasons to be able to make this statement.**

p.7 l.21 Why does the height of the tropopause affect the maximum height of the smoke layer? To first order, transport in the lower stratosphere is isentropic, so it is the tilt of isentropic surfaces that must be considered, not the tropopause (which crosses the isentropes).

**You are right, we removed the tropopause statement.**

p.7 l.22 'smoke reached 22 km'

**Improved!**

p.7 l.25-28 What is the point of the Sicard reference? It adds nothing to the argument

**We removed the sentence and the reference**

p.7 l.30 'from' not 'since'

**Improved!**

p.9 l.22 'introduction'

**We changed the text!**

p.22 l.1 omit 'shown

**Improved!**

p.22 l.6 'vertically average'

**Improved!**

[revised manuscript text omitted]

---

## Author Comment (AC2) · 26 Oct 2019

The reply letter is attached

Please also note the supplement to this comment:
https://www.atmos-chem-phys-discuss.net/acp-2019-615/acp-2019-615-AC2-supplement.pdf

———————————————————

---

## Referee Report (RR1)

Review of revised B19.

B19 have revised the text, some figures, and given greater advocacy to the Brewer Dobson Circulation (BDC) as the pathway for what they interpret as ascending aerosol structures observed in autumn 2017 over Mediterranean Europe. To their credit, B19 added more data to the analysis and depicted these data in a fuller manner than the original manuscript. They bolstered their BDC argument by referring to the now published Kloss et al. (2019) (K19) paper and invoking Jäger (2005), in which the impact of El Chichon and Pinatubo on a mid-latitude lidar station were emphasized. Jaeger implicated the BDC for lidar-based aerosol structures after the El Chichon and Pinatubo eruptions that resembled the ascending aerosol structures B19 report and discuss.

Unfortunately I found none of the BDC-related revisions convincing, and in fact concluded that the new content served to weaken the case for BDC influence in 2017. Considering that my original assessment was that this was a major issue, and that B19 have placed even greater advocacy on BDC in the revised manuscript, my assessment is that this is still a major concern that needs to be dealt with. B19 have responded satisfactorily to the additional concerns brought up by bother reviewers. Hence I would consider this paper worthy of publication after defensible revisions regarding the BDC attribution are made.

Below I elaborate on the BDC concern.

For reference, B19 state that meridional transport from the tropical stratospheric reservoir ensued n mid-September 2017. "However, during the autumn and winter season (from mid September to end of December) a northward transport of aerosols from the tropical stratospheric reservoir (TSR) towards the mid latitudes must be considered." This means that within about 1 month of the pyroCb injection (mid-August) in British Columbia (52°N), large abundances of stratospheric smoke completed a movement into the TSR and a subsequent movement to 32-35°N by way of the characteristically slow BDC. On its face this seems meteorologically improbable. As will be mentioned below, there is also no evidence of smoke at the required time and altitude observed in the TSR.

B19 have cited Kloss et al (K19) who claimed a BDC influence on tropical smoke they reported. Even if that point is conceded, K19's results are irreconcilable with those of B19. The only evidence K19 give for BDC-driven tropical ascent is a time series of vertically resolved aerosol data wherein the first hint of tropical stratospheric smoke is discerned in late October 2017. The images below, from B119 and K19, synchronized, illustrate the irreconcilability. The K19 aerosol ascent begins after the onset of perceived Mediterranean (extratropical) ascending structure (September 2017). Moreover, the K19 tropical layer never reaches the altitude of the B19 structure during the B19 time frame ("18-19 km to 22-23 km height from the beginning of October to the beginning of December"). Given the expected lag between the tropics and extratropics due to the characteristically slow BDC transport, there is no information available in K19 to support the argument that B19's earlier, higher structures come from the tropics.

The scenarios of Jäger (2005) and B19 are wholly different. The 2017 pyroCb eruption event was at 52°N; the smoke plume is entirely at high latitudes as late as mid-August. Even if it is acknowledged that some fraction of the pyroCb stratospheric smoke eventually got into the TSR, it is surely a small proportion of the source term (Bourassa et al., 2019). The Pintaubo and El Chichon aerosol source terms

were massive relative to the 2017 pyroCb event and exclusively tropical at source. Hence all meridional aerosol spread started from these massive source terms. Even so it took more than one month for the Pinatubo aerosol to arrive over central Europe (Jäger, 1992). It is noted here that the original Garmisch-Pinatubo timeline has been reinterpreted (Fromm et al., 2010) such that the earliest Pinatubo aerosols to reach Garmisch were~1 month later than reported by Jäger (1992), 2 months post eruption. According to Reiter et al. (1982) the onset of El Chichon aerosol at Garmisch was ~1 month after the eruption. So even under the relatively favorable situations of the two volcanoes, the impact on Europe lagged by one or more months from the injection date.

If indeed some of the Canadian pyroCb stratospheric smoke got into the TSR, then got lofted by the BDC, then got spread back to the extratropics, one might hypothesize that the residual concentration of aerosols would be much smaller than the "main" smoke over Europe transported there more directly. However, the particulate extinction plots in B19 do not suggest a systematic difference between these two populations. Hence it would have to be argued that the BDC-transported smoke would have a heritage of even greater concentrations (translated as extinction). Neither K19 nor any other published works on the Canadian pyroCb smoke expressly or implicitly show that the tropics contained such relatively large smoke concentrations.

The replaced figure in B19 on which they focus the BDC discussion (Figure 8) has a variety of structures. B19 acknowledge that fact. However, the newly constructed figure hinders their argumentation. There are very high layers earl on and what can be interpreted as descending structures within. With smoke aerosols at and above 22 km in September in the Mediterranean zone, it is difficult to sort out the reasons for all of the variations in altitude that are evident. Given the co-existence of structures with positive slope, negative slope, no slope, and even multiple simultaneous layers, attention to just the positive-slope structure seems to be artificially limited, in my opinion.

In my first review I described (but did not show) a back-trajectory analysis from a certain location within the suspected BDC-affected smoke structure, to see where it came from. The example I provided was from an Iberian aerosol layer observation not expressly reported in B19. Understandably, B19 did not comment on this suggestion. So with this review I chose two B19-reported observations from which to run back trajectories. They were chosen because they were both similarly high layers: ~22 km. One was from Limassol in the 2[nd] half of September. The other was from Evora in late October. The results, shown below, reveal paths almost 180° opposed. Neither indicates a path toward the tropics; both reveal a path into an area of observed extratropical smoke at the same altitude but outside the European sector analyzed by B19. The Limassol back trajectory was consistent with a stream of smoke reported by Bourassa et al. (2019). The Evora path also provided a direct connection with upstream smoke observations by CALIPSO (see the link in both examples). These connections provide fairly strong proof that a residual meridional circulation from the tropics was not in play and that direct, quasi-isoentropic extra-tropical dynamics was the more plausible pathway.

Given the above comments and analysis, I think it is exceedingly difficult to conclude that the BDC was responsible for the disposition of the apparently ascending structures reported by B19. Given the publication of K19 it is appropriate to acknowledge this preceding paper, and the potential for alternate

explanations. But the evidence leading to the firm conclusion of the primary role of BDC is not present, in my view.

**References**

Bourassa, A. E., Rieger, L. A., Zawada, D. J., Khaykin, S., Thomason, L. W., & Degenstein, D. A. (2019). Satellite limb observations of unprecedented forest
fire aerosol in the stratosphere. Journal of Geophysical Research: Atmospheres, 124, 9510–9519. https://doi.org/ 10.1029/2019JD030607

Jäger, H., 1992: The Pinatubo eruption cloud observed by lidar at Garmisch Partenkirchen. *Geophys. Res. Lett.,* **19**, 191–194.

Kloss, C., Berthet, G., Sellitto, P., Ploeger, F., Bucci, S., Khaykin, S., Jègou, F., Taha, G., Thomason, L.W., Barret, B., Le Flochmoen, E., von Hobe, M., Bossolasco, A., Bègue, N., and Legras, B.: Transport of the 2017 Canadian wildfire plume to the tropics and global stratosphere 35 via the Asian monsoon circulation, Atmos. Chem. Phys. Discuss., https://doi.org/10.5194/acp-2019-204, FINAL!!!...in review, 2019. ...accepted 7 Oct 2019

Reiter, R., H. Jäger, W. Carnuth, and W. Funk (1980), Lidar observations of the Mount St. Helens eruption clouds over mid-Europe, May to July 1980, *Geophys. Res. Lett.*, 7, 1099– 1101.

[Figure]

**K19 Figure 1D.**

**K19 Fig. 1D resized to match B19 Fig. 8.**

[Figure]

Onset of K19 stratospheric tropical smoke
~Last week of October

[Figure]

B19 Figure 8.

Onset of K19 stratospheric tropical smoke
Presumably there would be a lag before
any tropical material got to 30+N.

[Figure]

https://tinyurl.com/calipso-eastern-smoke

https://tinyurl.com/calipso-western-smoke

---

## Author Response (AR2)

Dear Editor!

These major revisions could be done very quickly. Again, we appreciate the comments of the reviewers and the time they used to help us to further improve the manuscript.

**Our answers in bold. We also indicate the new changes in the manuscript in BOLD.**
**The revised version of the paper is included in this letter (after the reply part)!**

Reviewer #2:

**We addressed all the five minor points mentioned by reviewer #2.**

1. In response to my comment about homogeneous and heterogeneous freezing in the original manuscript the authors have added: 'Homogeneous nucleation is the process in which droplets freeze (and no solid particle phase is involved). In the case of heterogeneous nucleation, a solid particle is needed to initiate the nucleation of an ice crystal, but can take place at much lower ice supersaturation as needed to iniate homogeneous freezing'. But the ice supersaturation is irrelevant to (immersion) freezing – i.e. when a solid particle is inside a liquid droplet the freezing process is simply a function of temperature (and the composition and structure of the nucleus). Later in the paper you do discuss deposition of ice which is indeed sensitive to supersaturation, but that is a different process to that described here. Note the typo too ('iniate').

**In the introduction, we are now a bit more precise and state that at cirrus level (e.g., -50 to -60°C) only deposition nucleation plays a role, so immersion freezing needs not to be discussed here...**

2. The last paragraph of the introduction seems out of place, and certainly the reference to the long-term record at Garmisch-Partenkirchen is irrelevant to this paper. Reference to the earlier work of Fromm is of course relevant, and I can see you wouldn't want to just move the text somewhere else, but it doesn't read well here.

**We removed the paragraph from the Introduction and mention the network activities (in 2001) now in Section 3.1 (bold part in page 7).**

3. P.10 l.19 remove brackets around Mamouri and Ansmann (cite not citep)

**Improved!**

4. P.12 l.27 I think you mean 'from the tropics to the extratropics'

**Improved!**

5. P.15. The new paragraph on INC is purely speculative and adds nothing to the paper. It should be removed. There is quite enough speculation in this paper already.

**We removed the paragraph. ... but still mention that cirrus studies in August-September 2017 would be interesting.**

Unfortunately I found none of the BDC-related revisions convincing, and in fact concluded that the new content served to weaken the case for BDC influence in 2017. Considering that my original assessment was that this was a major issue, and that B19 have placed even greater advocacy on BDC in the revised manuscript, my assessment is that this is still a major concern that needs to be dealt with. B19 have responded satisfactorily to the additional concerns brought up by bother reviewers. Hence I would consider this paper worthy of publication after defensible revisions regarding the BDC attribution are made.

For reference, B19 state that meridional transport from the tropical stratospheric reservoir ensued n mid-September 2017. "However, during the autumn and winter season (from mid September to end of December) a northward transport of aerosols from the tropical stratospheric reservoir (TSR) towards the mid latitudes must be considered." This means that within about 1 month of the pyroCb injection (mid-August) in British Columbia (52°N), large abundances of stratospheric smoke completed a movement into the TSR and a subsequent movement to 32-35°N by way of the characteristically slow BDC. On its face this seems meteorologically improbable. As will be mentioned below, there is also no evidence of smoke at the required time and altitude observed in the TSR.

B19 have cited Kloss et al (K19) who claimed a BDC influence on tropical smoke they reported. Even if that point is conceded, K19's results are irreconcilable with those of B19. The only evidence K19 give for BDC-driven tropical ascent is a time series of vertically resolved aerosol data wherein the first hint of tropical stratospheric smoke is discerned in late October 2017. The images below, from B119 and K19, synchronized, illustrate the irreconcilability. The K19 aerosol ascent begins after the onset of perceived Mediterranean (extratropical) ascending structure (September 2017). Moreover, the K19 tropical layer never reaches the altitude of the B19 structure during the B19 time frame ("18-19 km to 22-23 km height from the beginning of October to the beginning of December"). Given the expected lag between the tropics and extratropics due to the characteristically slow BDC transport, there is no information available in K19 to support the argument that B19's earlier, higher structures come from the tropics.

The scenarios of Jäger (2005) and B19 are wholly different. The 2017 pyroCb eruption event was at 52°N; the smoke plume is entirely at high latitudes as late as mid-August. Even if it is acknowledged that some fraction of the pyroCb stratospheric smoke eventually got into the TSR, it is surely a small proportion of the source term (Bourassa et al., 2019). The Pintaubo and El Chichon aerosol source terms were massive relative to the 2017 pyroCb event and exclusively tropical at source. Hence all meridional aerosol spread started from these massive source terms. Even so it took more than one month for the Pinatubo aerosol to arrive over central Europe (Jäger, 1992). It is noted here that the original Garmisch-Pinatubo timeline has been reinterpreted (Fromm et al., 2010) such that the earliest Pinatubo aerosols to reach Garmisch were~1 month later than reported by Jäger (1992), 2 months post eruption. According to Reiter et al. (1982) the onset of El Chichon aerosol at Garmisch was ~1 month after the eruption. So even under the relatively favorable situations of the two volcanoes, the impact on Europe lagged by one or more months from the injection date.

If indeed some of the Canadian pyroCb stratospheric smoke got into the TSR, then got lofted by the BDC, then got spread back to the extratropics, one might hypothesize that the residual concentration of aerosols would be much smaller than the "main" smoke over Europe transported there more directly. However, the particulate extinction plots in B19 do not suggest a systematic difference between these two populations. Hence it would have to be argued that the BDC-transported smoke would have a heritage of even greater concentrations (translated as extinction). Neither K19 nor any other published works on the Canadian pyroCb smoke expressly or implicitly show that the tropics contained such relatively large smoke concentrations.

The replaced figure in B19 on which they focus the BDC discussion (Figure 8) has a variety of structures. B19 acknowledge that fact. However, the newly constructed figure hinders their argumentation. There are very high layers earl on and what can be interpreted as descending structures within. With smoke aerosols at and above 22 km in September in the Mediterranean zone, it is difficult to sort out the reasons for all of the variations in altitude that are evident. Given the co-existence of structures with

positive slope, negative slope, no slope, and even multiple simultaneous layers, attention to just the positive-slope structure seems to be artificially limited, in my opinion.

In my first review I described (but did not show) a back-trajectory analysis from a certain location within the suspected BDC-affected smoke structure, to see where it came from. The example I provided was from an Iberian aerosol layer observation not expressly reported in B19. Understandably, B19 did not comment on this suggestion. So with this review I chose two B19-reported observations from which to run back trajectories. They were chosen because they were both similarly high layers: ~22 km. One was from Limassol in the 2nd half of September. The other was from Evora in late October. The results, shown below, reveal paths almost 180° opposed. Neither indicates a path toward the tropics; both reveal a path into an area of observed extratropical smoke at the same altitude but outside the European sector analyzed by B19. The Limassol back trajectory was consistent with a stream of smoke reported by Bourassa et al. (2019). The Evora path also provided a direct connection with upstream smoke observations by CALIPSO (see the link in both examples). These connections provide fairly strong proof that a residual meridional circulation from the tropics was not in play and that direct, quasiisoentropic extra-tropical dynamics was the more plausible pathway.

Given the above comments and analysis, I think it is exceedingly difficult to conclude that the BDC was responsible for the disposition of the apparently ascending structures reported by B19. Given the publication of K19 it is appropriate to acknowledge this preceding paper, and the potential for alternate explanations. But the evidence leading to the firm conclusion of the primary role of BDC is not present, in my view.

**We totally agree with the general (overall) opinion of reviewer #1 and revised the manuscript accordingly. We partly disagree regarding his comments on the comparison between lidar (b19) and satellite remote sensing products (K19). We will not comment on that in order to be short here and to save time.**

**The reviewer is right with his final statement: But the evidence leading to the firm conclusion of the primary role of BDC is not present, in my view.**

**Most of the arguments mentioned by reviewer #1 (outlined above) are justified and convincing and thus forced us to change the discussion! Thank You for the suggestions!**

**The main two points of criticism are:**

**(I)       The observations do not CONFIRM a leading influence of BDC in autumn and winter 2017 on the aerosol transport, layering, and observed ascending features.**

**(II)      The aerosol potentially transported from the tropics to the extra tropics may not be the Canadian fire smoke. The aerosol mixture is simply unknown and it is almost unlikely that smoke contributed.**

**Now to our response leading to the made changes (in bold in the manuscript).**

**The FACTS are still as follows:**

**A section on BDC is required! We cannot skip and thus ignore such a potential (possible) impact in the discussion. We cannot ignore an effect that may have had an impact.**

**The BDC moves aerosol from the tropical reservoir out of the tropics.  BDC simply works in the northern hemisphere in each winter halfyear!  And this fact needs to be discussed.**

**That BDC worked also during the winter 2017-2018  is documented and thus 'confirmed' by Kloss et al. (2019).**

**But the reviewer is right: We need to avoid the impression that our observations clearly indicate that**

the BDC was dominating (played the primary role). This is misleading! Therefore we clearly state now: This is just an option! (see Sect.3.3.1, page 12). Nevertheless, we explain how the BDC works and what the effects are. And this is nicely presented by Jäger (2005). And we also leave in the Kloss et al. (2019) findings that the BDC was 'visible' in the smoke data (October 2017 to March 2018). These are just the facts.

Finally: Regarding the type of aerosol that was moved from the tropical stratospheric reservoir northwards: We leave it now open! We state…. It may be smoke, or tropical volcanic particles, or even anthropogenic pollution or dust…  entering the stratosphere in the upwelling branch of the BDC… (page 12)!

To be consequent, we changed the Abstract and the Conclusions section! We do no longer even mention the  BDC and the possible impact in the Abstract and Conclusions section. This was clearly misleading, and thus a mistake! The discussion of transport and lifting mechanisms is just a minor part of the paper.

In conclusion: We got the message of the reviewer and removed the impression (primary role of BDC) totally from the paper.

[revised manuscript text omitted]